# S2R-HDR: A Large-Scale Rendered Dataset for HDR Fusion

**Yujin Wang[1]\*, Jiarui Wu[2,1]\*, Yichen Bian[1,4]\*, Fan Zhang[1], Tianfan Xue[2,1,3]**

[1]Shanghai AI Laboratory,[2]CUHK MMLab,[3]CPII under InnoHK,[4]Shanghai Jiao Tong University

{wangyujin, bianyichen, zhangfan}@pjlab.org.cn,
{wj024, tfxue}@ie.cuhk.edu.hk

## Abstract

The generalization of learning-based high dynamic range (HDR) fusion is often limited by the availability of training data, as collecting large-scale HDR images from dynamic scenes is both costly and technically challenging. To address these challenges, we propose S2R-HDR, the first large-scale high-quality synthetic dataset for HDR fusion, with 24,000 HDR samples. Using Unreal Engine 5, we design a diverse set of realistic HDR scenes that encompass various dynamic elements, motion types, high dynamic range scenes, and lighting. Additionally, we develop an efficient rendering pipeline to generate realistic HDR images. To further mitigate the domain gap between synthetic and real-world data, we introduce S2R-Adapter, a domain adaptation designed to bridge this gap and enhance the generalization ability of models. Experimental results on real-world datasets demonstrate that our approach achieves state-of-the-art HDR fusion performance. Dataset and code are available at `https://openimaginglab.github.io/S2R-HDR`.

## 1 Introduction

High dynamic range (HDR) fusion plays a crucial role in various real-world applications, such as computational photography, visual perception, and autonomous driving. Despite notable advancements in HDR fusion techniques (Yan et al., 2019; Kalantari & Ramamoorthi, 2017; Liu et al., 2022; Tel et al., 2023; Kong et al., 2024) in recent years, models trained on small-scale datasets (Kalantari & Ramamoorthi, 2017; Chen et al., 2021; Kong et al., 2024; Tel et al., 2023) still face limitations in generalizing to complex scenes. Additionally, due to limited data scale, the complexity and challenges of HDR fusion have yet to be fully explored, particularly in scenarios involving large motion and direct sunlight, as illustrated in Figure 1.

In real-world scenarios, collecting comprehensive, high-quality large-scale HDR datasets for dynamic scenes is time-consuming, resource-intensive, and poses significant technical challenges. Uncontrollable elements such as lighting conditions, weather variations, and dynamic objects like animals and vehicles make it difficult to fully control the data acquisition process. Capturing extreme high dynamic range scenarios—such as environments with direct sunlight—poses an even greater challenge, like Figure 1. Consequently, existing HDR datasets (Kalantari & Ramamoorthi, 2017; Tel et al., 2023; Kong et al., 2024; Chen et al., 2021; Shu et al., 2024) are generally limited to artificially controlled dynamic scenes and fail to capture the diversity of real-world environments. For example, some datasets focus exclusively on human motion, overlooking other essential dynamic elements, such as animals and vehicles. Moreover, existing HDR datasets with ground truth fusion results are typically small. For instance, Kong et al. (2024) built the latest dataset with 123 samples. Models trained on these small datasets are prone to overfitting, limiting their performance under challenging scenarios. A larger synthetic dataset (Barua et al., 2025) has been proposed recently for the single-image LDR to HDR conversion task, which is intrinsically limited by the absence of complementary exposure information, resulting in unrecoverable details in saturated regions.

---

*Equal contribution.

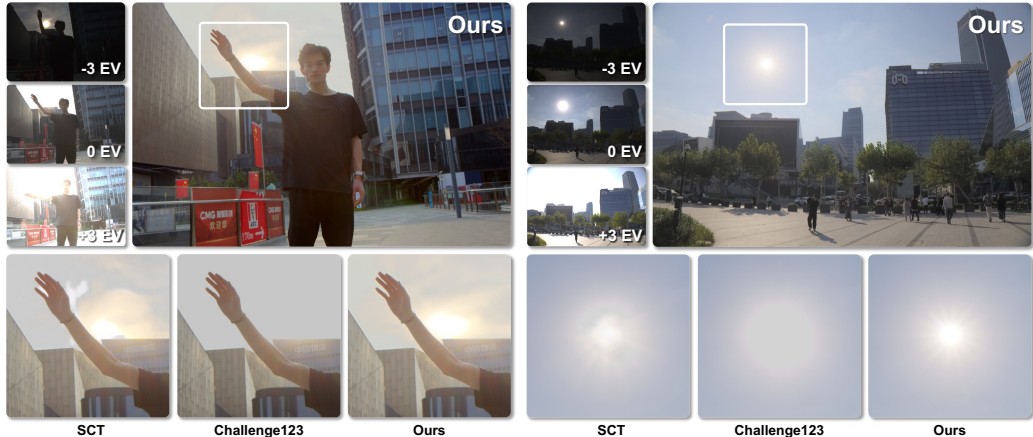

Figure 1: Comparing HDR fusion models (Kong et al., 2024) trained on our S2R-HDR dataset, with the proposed domain adapter S2R-Adapter, with the same model trained on previous SCT (Tel et al., 2023) and Challenge123 (Kong et al., 2024) datasets. Results show our dataset and training scheme can reduce ghosting artifacts under large motion (left) and recover very high dynamic range scenes, such as direct sunlight (right).

To address these limitations, we introduce *S2R-HDR*, the first large-scale HDR synthetic dataset designed for HDR fusion. S2R-HDR features several distinctive characteristics: 1) *High Quality*: Inspired by prior works (Li et al., 2023; Yang et al., 2023; Hu et al., 2023; Chen et al., 2023; Yin et al., 2024), we render high-quality raw HDR data using Unreal Engine, with realistic lighting, shadow, weather, and motion effects. 2) *Large Scale*: The dataset contains 24,000 HDR images, around 166 times larger than typical datasets (Kalantari & Ramamoorthi, 2017; Tel et al., 2023; Kong et al., 2024). 3) *Diversity*: The dataset encompasses different motion types and lighting. It also covers different dynamic elements such as animals, humans, and vehicles across a variety of indoor and outdoor settings. 4) *Controllable Environment*: Using tools developed based on xrfeitoria (Contributors, 2023), we can flexibly control environmental factors to create diverse data.

While rendering engines can generate a large volume of high-quality synthetic data, a domain gap exists between synthetic and real data, particularly in texture distribution, as discussed in Appendix A.2. To address this, we propose S2R-Adapter, a plug-and-play simulation-to-real domain adaptation approach designed to bridge this gap. This approach can be applied to both labeled and unlabeled data, meaning even if the target real HDR datasets do not have the ground truth fusion result, we can still adapt to it. To achieve this, inspired by previous works (Hu et al., 2021; Yang et al., 2024; Liu et al., 2023a), our S2R-Adapter consists of two branches: 1) A *share branch* manages knowledge sharing, which ensures the knowledge learned from synthetic data are not forgotten, and 2) a *transfer branch* facilitates knowledge transfer, which ensures the model can adapt to real input.

Additionally, our training strategy can be applied to different network structures, including both CNN-based and transformer-based models. Integrating this strategy using re-reparameterization (Ding et al., 2021) incurs no extra computational overhead during inference.

Experimental results on both labeled and unlabeled real datasets demonstrate that the proposed dataset and method significantly enhance the performance of HDR fusion models trained on synthetic data when applied to real scenes, achieving state-of-the-art results. Our study not only provides a new solution for HDR fusion but also presents a feasible path for generalization in fields where data acquisition is challenging.

## 2 RELATED WORKS

**Image HDR datasets.** Datasets are essential for the development and evaluation of algorithms. Before the deep learning era, Sen et al. (2012) and Tursun et al. (2016) provided real-world HDR datasets containing 8 and 16 scenes, respectively, and Kalantari & Ramamoorthi (2017) further introduced the first paired LDR-HDR dataset with 89 pairs. Prabhakar et al. (2019) later expanded

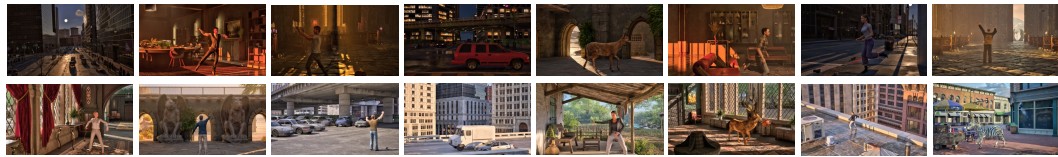

Figure 2: Illustration of our S2R-HDR dataset, covering both indoor and outdoor environments under diverse lighting conditions, including daytime, dusk, and nighttime, as well as various motion types such as humans, animals, and vehicles.

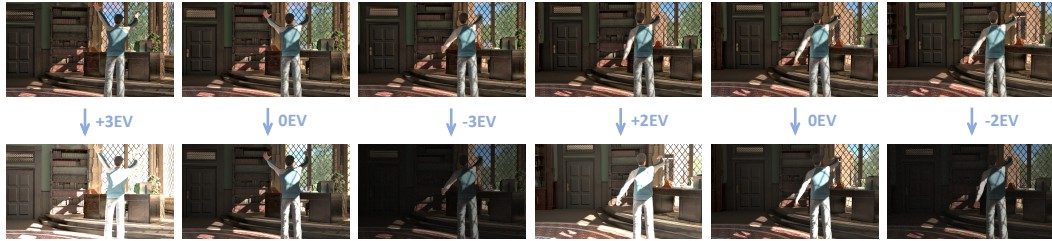

Figure 3: Visualization of our sequence data and synthesized multi-exposure LDR images. Since the dataset consists of raw HDR sequences, it enables effortless data augmentation, such as brightness enhancement and motion amplitude adjustment.

this to 582 LDR-HDR pairs and Tel et al. (2023) collected a dataset focusing on foreground objects and larger motion variations, with 144 samples. Other datasets are also built for deghosting (Shu et al., 2024), mobile imaging (Liu et al., 2023b) , or large motion (Kong et al., 2024).

**Image HDR methods.** Deep learning has been introduced into the field of HDR fusion due to its remarkable performance in image processing. Early researchers designed an alignment and fusion pipeline (Kalantari & Ramamoorthi, 2017; Wu et al., 2018). Subsequent works (Catley-Chandar et al., 2022; Chung & Cho, 2023; Liu et al., 2021; Yan et al., 2023a) focused on improving the alignment process by developing more advanced modules to handle motion artifacts across different exposures. Kong et al. (2024) also proposed a novel efficient processing network.

Over time, several alternative pipelines for HDR fusion have been proposed, using attention mechanisms (Yan et al., 2019), non-local blocks (Yan et al., 2020), generative adversarial networks (Niu et al., 2021), or multi-step fusion (Ye et al., 2021). Recently, transformer models have shown promising results in HDR fusion (Song et al., 2022; Liu et al., 2022). Tel et al. (2023) also developed a semantic-consistent, alignment-free transformer for HDR fusion. Recently, diffusion models have also been introduced to HDR fusion by (Yan et al., 2023b; Hu et al., 2024; Chen et al., 2025), further improving the performance. Self-supervised approaches have also been introduced to HDR fusion, with the SelfHDR method (Zhang et al., 2024) utilizing a color and structure-focused network to effectively handle deghosting.

**Sim-to-real domain adaptation.** Domain adaptation has been widely used to transfer models trained on synthetic data to real-world settings. To address the domain shifts, researchers use either adversarial approaches (Ganin et al., 2016; Tzeng et al., 2017) or domain randomization (Tobin et al., 2017). Recently, adapter-based domain adaptation (Hu et al., 2021; Chen et al., 2022; Sung et al., 2022) has been proven to be more effective. Adapters (Hu et al., 2021; Chen et al., 2022) are a form of parameter-efficient fine-tuning (PEFT) (Hu et al., 2021; Zaken et al., 2021; Gao et al., 2021; Hu et al., 2022a), which require fewer parameters than full retraining and help mitigate catastrophic forgetting (Chen et al., 2022; Liu et al., 2023a) in domain adaptation. Additionally, Test-Time Adaptation (TTA) (Kundu et al., 2020; Boudiaf et al., 2023; Wang et al., 2022; Liu et al., 2023a; Chen et al., 2022) has been extensively explored, aiming to adapt a pre-trained model to unknown target domains during test-time, without any labeled or source domain data.

## 3 S2R-HDR DATASET

Previously, to create an HDR dataset with ground truth, researchers often use a beam splitter and two cameras to simultaneously capture images with two different exposures (Froehlich et al., 2014;

Table 1: Qualitative comparison and analysis of different HDR datasets. Besides the DR, all numbers are in percentage.

| Dataset | Extent of HDR | | Intra-frame Diversity | | | Overall Style | | Size |
| --- | --- | --- | --- | --- | --- | --- | --- | --- |
| | FHLP ↑ | EHL ↑ | SI ↑ | CF ↑ | stdL ↑ | ALL ↑ | DR ↑ | |
| Kalantari (Kalantari & Ramamoorthi, 2017) | 15.07 | 3.07 | 18.4 | 4.74 | 10.02 | 6.19 | 2.71 | 89 |
| SCT (Tel et al., 2023) | 12.43 | 2.43 | 18.25 | 3.92 | 9.39 | 5.44 | 2.55 | 144 |
| Challenge123 (Kong et al., 2024) | 26.91 | 5.19 | 20.47 | 5.19 | 12.73 | 9.88 | 2.36 | 123 |
| S2R-HDR | **28.02** | **5.47** | **38.02** | **14.96** | **15.16** | **10.53** | **3.86** | **24000** |

Wang et al., 2021). The beam splitter only has two different exposures, which limits the dynamic range of the image. However, there are various high dynamic range scenarios in natural scenes, such as environments with direct sunlight. Accurately extracting tens of thousands of data samples from these scenes is a significant challenge. Previously, the largest commonly used dataset contained only 144 images (Tel et al., 2023), whereas ours includes 24,000 HDR images, representing a substantial leap in scale and diversity.

Moreover, capturing the ground truth often requires capturing different exposure images frame-by-frame (Kalantari & Ramamoorthi, 2017; Tel et al., 2023; Kong et al., 2024; Chen et al., 2021; Shu et al., 2024) and manually controlling motion between frames, making capturing extremely time-consuming. The captured motions are often limited and unrealistic, most of them are just basic human movements. These limitations have made it difficult to scale HDR datasets both in terms of size and motion variety. Below, we discuss how we solve all these challenges.

## 3.1 RENDERING DESIGN

Rendering high-quality HDR data presents several challenges. One challenge is that rendered images have a different distribution compared to the actual raw sensor data captured by cameras. To mitigate this difference, we made several improvements. First, by default, rendered images have a baked-in tone mapping, an irreversible process that compresses dynamic range for standard displays, making it hard to recover original HDR data. To overcome this, we design a custom UE5 (Unreal Engine 5) rendering pipeline that modifies tone mapping and gamma correction, ensuring the output remains in linear HDR space, and stores results in floating-point formats (EXR) to prevent data quantization. This approach ensures greater accuracy and makes the rendered data more suitable for HDR-related tasks. Second, we also simulate imperfections during handheld capturing. We incorporated camera shake simulation into our camera pose control to replicate the vibrations and instabilities that occur during real-world capture. This ensures that the rendered data closely mimics real-world shooting conditions, yielding more realistic HDR data for image processing and model training.

Another challenge is to construct realistic and diversified HDR scenes, with varying motion, lighting, and environmental details. To tackle this, we design and curate a diverse range of dynamic scene materials, including common moving objects such as animals, pedestrians, and vehicles, ensuring that the scenes exhibit a high degree of dynamism and complexity, as shown in Figure 2. Additionally, we carefully build a variety of high dynamic range scenes, encompassing both indoor and outdoor environments, various lighting conditions across different times of day, and extreme lighting scenarios. This diversity ensures that the generated HDR data simulates a broad range of real-world environments as much as possible. Additional examples of our motion materials and HDR scenes can be found in Appendix B.1 and Appendix B.2.

In total, we rendered 1,000 sequences, each containing 24 frames, resulting in a dataset of 24,000 HDR images, all stored in EXR format at a resolution of $1920 \times 1080$. As demonstrated in Figure 2, our rendered data encompasses a variety of environments and includes a broad range of motion types, showcasing a high degree of variability. Furthermore, since the data is in linear HDR format, it facilitates flexible data augmentation, enabling the easy generation of different LDR (low dynamic range) images, as shown in Figure 3.

## 3.2 STATISTICS AND ANALYSIS

We further analyze the diversity of S2R-HDR in comparison to previous datasets (Kalantari & Ramamoorthi, 2017; Tel et al., 2023; Kong et al., 2024). Following the methodology of Shu et al.

Figure 4: Structure of S2R-Adapter and t-SNE visualization of feature representations.

(2024); Guo et al. (2023); Hu et al. (2022b), we use seven metrics to evaluate the diversity of different datasets across three dimensions: the extent of HDR, intra-frame diversity, and overall HDR style. As shown in Table 1, the S2R-HDR dataset outperforms all prior datasets across these metrics. The "Extent of HDR" metric demonstrates that our dataset covers a broader range of highlights, indicating an extended highlight range. The "Intra-frame Diversity" metric suggests that our images contain more detailed information and richer content. Finally, the "Overall Style" metric reveals that S2R-HDR exhibits a significantly higher dynamic range, surpassing the performance of previous datasets. Details of seven metrics can be found in Appendix B.4.

Additionally, to visually illustrate the distribution between our dataset and existing real-world datasets (Kalantari & Ramamoorthi, 2017; Tel et al., 2023; Kong et al., 2024), we extract seven-dimensional feature vectors for each image and apply t-SNE (Van der Maaten & Hinton, 2008) for dimensionality reduction. As shown in Figure 8 (detailed in Appendix A.2), our S2R-HDR dataset spans a broader range in terms of data diversity. Additional data samples, along with optical flow, depth, and normal maps, are provided in Appendix B.6 and Appendix B.7, where we also discuss further application scenarios.

## 4 DOMAIN ADAPTION

With all the careful design proposed in the previous section, there is still a noticeable gap between the synthetic S2R-HDR dataset and the real one, as shown in the t-SNE visualization in Appendix A.2. Thus, it is crucial to adapt the model trained on a large-scale rendered dataset to a small-scale real one. Still, direct fine-tuning on labeled real data can lead to overfitting and knowledge forgetting (Yosinski et al., 2014; Kirkpatrick et al., 2017).

To mitigate knowledge forgetting, we propose S2R-Adapter, visual adapters designed specifically for the HDR Fusion task, which enhance knowledge control.

This is inspired by recent studies (Liu et al., 2023a; Chen et al., 2022; Sung et al., 2022), which suggest that adapters (Hu et al., 2021; Rebuffi et al., 2017; Chen et al., 2022) can mitigate forgetting in high-level vision tasks. Our adapter consists of two branches: a *share branch* to preserve shared knowledge from the rendered dataset, and a *transfer branch* to learn domain-specific knowledge from the real dataset, as shown in Figure 4 (a). We chose this design because we want to utilize both the shared knowledge from S2R-HDR to address large motion and dynamic range fusion, and the domain-specific knowledge from the real dataset, like more realistic textures.

More specifically, the proposed S2R-Adapter uses a plug-and-play structure, which can be attached to any pre-trained layers performing matrix multiplication (e.g., Linear Layer, Convolution Layer). Following Liu et al. (2023a), we use a low-rank adapter as the share branch, which can better address knowledge forgetting, and use a high-rank adapter as the transfer branch, which can better extract domain-specific knowledge. Below we introduce details of each branch.

**Shared branch.** Considering a linear layer. Let the pre-trained weight matrix be $W_0 \in \mathbb{R}^{h_{out} \times h_{in}}$, with input feature $x$. The original output of this layer is $W_0 x$. The shared branch uses a low-rank adapter, projecting the feature with a down-projection matrix $V_s \in \mathbb{R}^{h_{in} \times r_s}$, followed by an up-projection matrix $U_s \in \mathbb{R}^{r_s \times h_{out}}$, where the rank $r_s \ll \min(h_{in}, h_{out})$. The output of the shared branch is $f_s = U_s V_s x$.

**Transfer branch.** The transfer branch employs a high-rank adapter structure, starting with an up-projection matrix $V_t \in \mathbb{R}^{h_{in} \times r_t}$, followed by a down-projection matrix $U_t \in \mathbb{R}^{r_t \times h_{out}}$, where the rank $r_t \geq \max(h_{in}, h_{out})$. Thus, the output of the transfer branch is $f_t = U_t V_t x$.

The output features of the two branches are scaled by two separate factors $\alpha_s, \alpha_t$, then added to the pre-trained weight output:

$$f = W_0 x + \alpha_s \times f_s + \alpha_t \times f_t. \tag{1}$$

The scale factors $\alpha_s$ and $\alpha_t$ control the trade-off between the shared knowledge and the transfer to the real domain distribution.

**Verification using t-SNE.** To verify the effectiveness of the proposed share branch and transfer branch adapters, we visualize the distributions of the rendered and real images using t-SNE (Van der Maaten & Hinton, 2008) in Figure 4 (b). From the share branch adapter, the feature distributions are consistent between the real and rendered domain, indicating that the share branch can ignore the domain difference between the real and the rendered domain, preserving the shared knowledge from forgetting. On the other hand, the transfer branch better separates the real distribution from the rendered distribution, showing its capability to model the real data distribution better and extract domain-specific knowledge in the real domain.

**Training with labeled data.** In this study, we consider two domain adaptation tasks. One is adapting to real domains *with ground-truth* labels. The other is generalizing to any real domains *without ground-truth* labels during inference. When the labeled real domain data is available, we inject our S2R-Adapter into the pre-trained model and fine-tune the system on the labeled data. We also learn the scale factors $\alpha_s$ and $\alpha_t$ to ensure the optimal trade-off between shared knowledge and transferred knowledge on the real domain distribution.

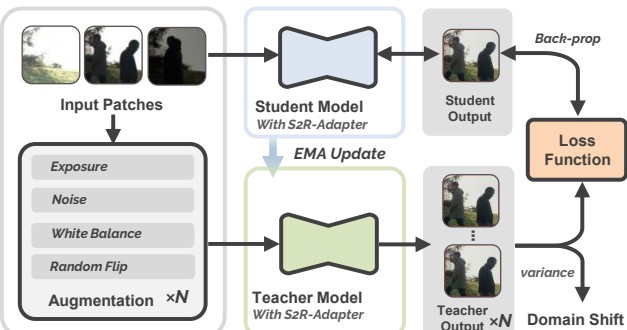

Figure 5: S2R-Adapter Framework on test-time adaptation without ground-truth data.

**Test-time adaptation with unlabeled data.** During test-time adaptation, no labeled real domain data is available, and each sample is seen only once. Therefore, $\alpha_s$ and $\alpha_t$ cannot be learned across the real domain. Moreover, each test sample's varying distance to the rendered domain requires adaptive scaling of transfer and shared branches. Therefore, inspired by Liu et al. (2023a); Ovadia et al. (2019), we dynamically adjust the scale factors using domain shift. For larger shifts, we increase the transfer branch's scale factor, encouraging more knowledge from the real domain. For smaller shifts, we allocate more from the shared branch, preserving rendered domain knowledge. Domain shift is measured by model uncertainty, following Wang et al. (2022); Liu et al. (2023a); Roy et al. (2022); Ovadia et al. (2019). In our HDR Fusion task, we augment input samples $N$ times and calculate variance across $N$ outputs as the uncertainty value $\mathcal{U}(x)$. Augmentations include adjusting exposure, white balance, noise levels, and random flips. With the uncertainty value, we adaptively adjust scale factors:

$$\alpha_s = 1 - \mathcal{U}(x); \quad \alpha_t = 1 + \mathcal{U}(x). \tag{2}$$

Following previous works on test-time adaptation (Yang et al., 2024; Wang et al., 2022), we utilize the mean-teacher framework. As shown in Figure 5, we inject S2R-Adapters to both the teacher model $\mathcal{T}$ and the student model $\mathcal{S}$. We initialized both models with pre-trained weights on the rendered domain. Following Liu et al. (2023a), the teacher model generates uncertainty values and pseudo-labels $\widetilde{y}$ for updating the S2R-Adapters. The student model is optimized by the loss between

Table 2: Experimental results on SCT (Tel et al., 2023) and Challenge123 (Kong et al., 2024) dataset with ground-truth. We first trained the two baseline networks on the S2R-HDR dataset, followed by simulation-to-real domain adaptation on the SCT and Challenge123 training sets using the S2R-Adapter. In contrast, the other methods were directly trained on the SCT and Challenge123 training sets. The results marked with * are those recalculated using images provided by Tel et al. (2023).

| Methods | Train/Adaptation/Test on SCT (Tel et al., 2023) | | | | | Train/Adaptation/Test on Challenge123 (Kong et al., 2024) | | | | |
|---|---|---|---|---|---|---|---|---|---|---|
| | PSNR-$\mu$ | PSNR-$\ell$ | SSIM-$\mu$ | SSIM-$\ell$ | HDR-VDP2* | PSNR-$\mu$ | PSNR-$\ell$ | SSIM-$\mu$ | SSIM-$\ell$ | HDR-VDP2 |
| NHDRRNet (Yan et al., 2020) | 36.68 | 39.61 | 0.9590 | 0.9853 | 63.72 | 37.82 | 26.75 | 0.9769 | 0.9632 | 53.38 |
| DHDRNet (Kalantari & Ramamoorthi, 2017) | 40.05 | 43.37 | 0.9794 | 0.9924 | 65.50 | 37.83 | 29.62 | 0.9707 | 0.9705 | 51.32 |
| AHDRNet (Yan et al., 2019) | 42.08 | 45.30 | 0.9837 | 0.9943 | 67.30 | 40.44 | 28.13 | 0.9877 | 0.9703 | 54.58 |
| DiffHDR (Yan et al., 2023b) | 42.77 | 47.11 | 0.9854 | 0.9957 | 69.43 | 38.78 | 26.85 | 0.9890 | 0.9745 | 53.38 |
| HDR-Transformer (Liu et al., 2022) | 42.39 | 46.35 | 0.9844 | 0.9948 | 67.73 | 40.70 | 28.72 | 0.9881 | 0.9731 | 54.63 |
| SCTNet (Tel et al., 2023) | 42.55 | 47.51 | 0.9850 | 0.9952 | 69.22 | 40.65 | 28.73 | 0.9882 | 0.9721 | 54.35 |
| SCTNet w S2R (Ours) | 43.24 | 48.32 | 0.9872 | 0.9962 | 69.33 | 42.58 | 30.68 | 0.9915 | 0.9805 | 55.35 |
| SAFNet (Kong et al., 2024) | 42.66 | 48.38 | 0.9831 | 0.9955 | 68.78 | 41.88 | 29.73 | 0.9897 | 0.9784 | 55.07 |
| SAFNet w S2R (Ours) | 43.33 | 48.90 | 0.9864 | 0.9959 | 70.00 | 43.43 | 31.84 | 0.9915 | 0.9824 | 56.51 |

the student output $\hat{y}$ and the pseudo-label $\widetilde{y}$. The teacher model updates via the exponential moving average (EMA) of the student model:

$$\mathcal{T}^t = \lambda \mathcal{T}^{t-1} + (1 - \lambda)\mathcal{S}^t, \tag{3}$$

where $t$ is the test step, $\lambda$ is set to 0.999, following Tarvainen & Valpola (2017).

## 5 EXPERIMENTS

**Datasets.** In line with the latest research (Tel et al., 2023; Kong et al., 2024), we train and evaluate our models on recent HDR datasets: the SCT Dataset(Tel et al., 2023), which contains 108 training samples and 36 test samples featuring dynamic scenes with significant foreground or camera motion; and the Challenge123 Dataset(Kong et al., 2024), a complex multi-exposure HDR dataset collected using a vivo X90 Pro+, comprising 96 training samples and 27 test samples.

**Experiment details.** We select the three latest methods (Liu et al., 2022; Tel et al., 2023; Kong et al., 2024) as our baselines: HDR-Transformer (Liu et al., 2022) and SCTNet (Tel et al., 2023) are transformer-based approaches, while SAFNet (Kong et al., 2024) is a CNN-based approach. When training these methods on our S2R-HDR dataset, we first generate three different exposure LDR images from the original HDR images. Then, we apply the same data augmentation and training strategy. Regarding adaptation steps, for supervised adaptation with ground-truth labels, we fine-tune the model for about 30 epochs. For the self-supervised mode (test-time adaptation without ground-truth), the model adapts as it processes the data during test time, so each sample is seen only once.

**Evaluation metrics.** We employ commonly used metrics, including PSNR and SSIM, along with HDR-VDP2 (Mantiuk et al., 2011), a metric designed for HDR evaluation. PSNR and SSIM are computed in both linear and $\mu$-law tone-mapped domains, denoted as $-\ell$ and $-\mu$, respectively.

### 5.1 RESULTS

**Results on test datasets with ground truth.** To validate the effectiveness of our method (S2R-HDR dataset and S2R Adapter), we conducted a comparative study on the latest SCT (Tel et al., 2023) and Challenge123 (Kong et al., 2024) datasets against seven widely adopted HDR approaches, including both CNN-based (Kalantari & Ramamoorthi, 2017; Yan et al., 2020; 2019; Tel et al., 2023), Transformer-based (Liu et al., 2022; Tel et al., 2023) and diffusion-based (Yan et al., 2023b) models. We selected the latest SCTNet and SAFNet as our baseline networks, where SCTNet represents the Transformer-based method and SAFNet represents the CNN-based method. Specifically, we first trained the two baseline networks on the S2R-HDR dataset, followed by synthetic-to-real domain adaptation on the SCT and Challenge123 training sets using the S2R Adapter. In contrast, the other methods were directly trained on the SCT and Challenge123 training sets.

As shown in Table 2, our method achieved the best results on both datasets. In terms of the PSNR-$\mu$ metric, our approach demonstrated at least a 0.6dB improvement over both baseline networks on PSNR-$\mu$, and notably achieved a significant 2dB gain on the Challenge123 dataset across both baselines. Additionally, we provide a comparative analysis of visual effects, as illustrated in Figure 6.

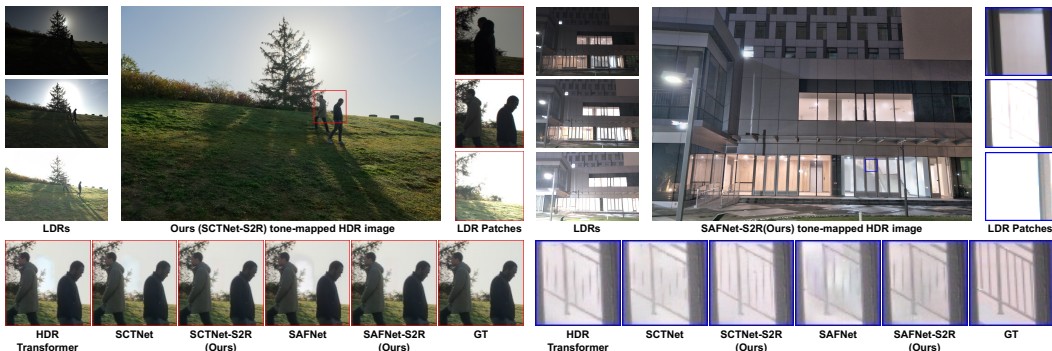

Figure 6: Visual results on the SCT (Tel et al., 2023) datasets (left) and Challenge123 (Kong et al., 2024) datasets (right) with ground-truth training data. Our method effectively eliminates artifacts caused by motion occlusions, delivering superior visual quality.

Table 3: Experimental result on SCT (Tel et al., 2023) and Challenge123 (Kong et al., 2024) without ground-truth (test-time adaptation). We report the testing results of baselines pre-trained on real-world datasets generalizing to the SCT and Challenge123 test datasets, followed by the S2R-Adapter test-time adaptation results of SCTNet and SAFNet pre-trained on S2R-HDR.

| Methods | | Test on SCT (Tel et al., 2023) | | | | | Test on Challenge123 (Kong et al., 2024) | | | | | |
|---|---|---|---|---|---|---|---|---|---|---|---|---|
| | Train | PSNR-$\mu$ | PSNR-$\ell$ | SSIM-$\mu$ | SSIM-$\ell$ | HDR-VDP2 | Train | PSNR-$\mu$ | PSNR-$\ell$ | SSIM-$\mu$ | SSIM-$\ell$ | DHR-VDP2 |
| DiffHDR (Yan et al., 2023b) | Challenge123 | 32.33 | 35.35 | 0.9497 | 0.9582 | 64.16 | SCT | 34.59 | 25.33 | 0.9748 | 0.9603 | 52.83 |
| HDR-Transformer (Liu et al., 2022) | | 31.94 | 34.23 | 0.9518 | 0.9503 | 62.70 | | 34.48 | 24.60 | 0.9744 | 0.9573 | 52.69 |
| SCTNet (Tel et al., 2023) | | 32.60 | 35.93 | 0.9535 | 0.9639 | 63.50 | | 34.57 | 25.07 | 0.9753 | 0.9599 | 52.09 |
| SAFNet (Kong et al., 2024) | | 35.14 | 38.77 | 0.9619 | 0.9868 | 64.03 | | 34.26 | 25.50 | 0.9718 | 0.9590 | 52.69 |
| SCTNet | S2R-HDR | 34.83 | 42.32 | 0.9526 | 0.9933 | 66.69 | S2R-HDR | 41.49 | 30.37 | 0.9862 | 0.9796 | 55.75 |
| SCTNet w S2R-Adapter (Ours) | | **35.35** | **43.33** | **0.9563** | **0.9936** | **67.84** | | **41.71** | **30.39** | **0.9876** | **0.9797** | **55.84** |
| SAFNet | S2R-HDR | 34.89 | 43.85 | 0.9500 | 0.9939 | 68.12 | S2R-HDR | 42.75 | 32.11 | 0.9872 | 0.9822 | **57.52** |
| SAFNet w S2R-Adapter (Ours) | | **36.28** | **47.23** | **0.9586** | **0.9949** | **68.40** | | **43.01** | **32.29** | **0.9884** | **0.9831** | 57.38 |

Our method effectively reduces artifacts caused by motion occlusions, delivering superior visual quality. We further visualize the difference maps of model output on the SCT dataset before and after applying the S2R-Adapter in Appendix A.2. The results show that our S2R-Adapter effectively reduces the domain gap, especially in texture-rich areas.

**Results on test datasets without ground truth.** We conduct the following experiment to validate the effectiveness of S2R-Adapter when generalizing to unseen test datasets where ground truth labels are not available for the adapted models. The pre-trained models are adapted to unseen target datasets SCT (Tel et al., 2023) and Challenge123 (Kong et al., 2024), without accessing the ground truth of the target domain during adaptation. Models will see each test sample only once. As shown in Table 3, compared with SCTNet and SAFNet trained on existing real-world datasets, models trained on our S2R-HDR dataset coupled with our S2R-Adapter can more effectively generalize to the unseen target domain. For instance, using SAFNet on the SCT dataset, our approach achieved a 1.1dB improvement in PSNR-$\mu$ and an 8.46dB improvement in PSNR-$\ell$ compared to the best baselines. The S2R-Adapter alone provides 1.39dB and 3.38dB improvements in PSNR-$\mu$ and PSNR-$\ell$, respectively.

With our test-time adaptation framework, models pre-trained on our dataset can effectively generalize to unseen images. A qualitative comparison using real-captured data without ground truth is illustrated in Figure 7. Our method effectively alleviates artifacts in highlight areas during nighttime and reduces ghosting caused by large motions. More visual comparisons are available in Appendix A.4.

## 5.2 ABLATION STUDY

**Effectiveness of S2R-HDR dataset.** To evaluate the effectiveness of the S2R-HDR dataset, we select the three latest methods (Liu et al., 2022; Tel et al., 2023; Kong et al., 2024) as baselines, which include both transformer-based and CNN-based approaches. Additionally, we chose the two most recent datasets, SCT (Tel et al., 2023) and Challenge123 (Kong et al., 2024), as comparative datasets.

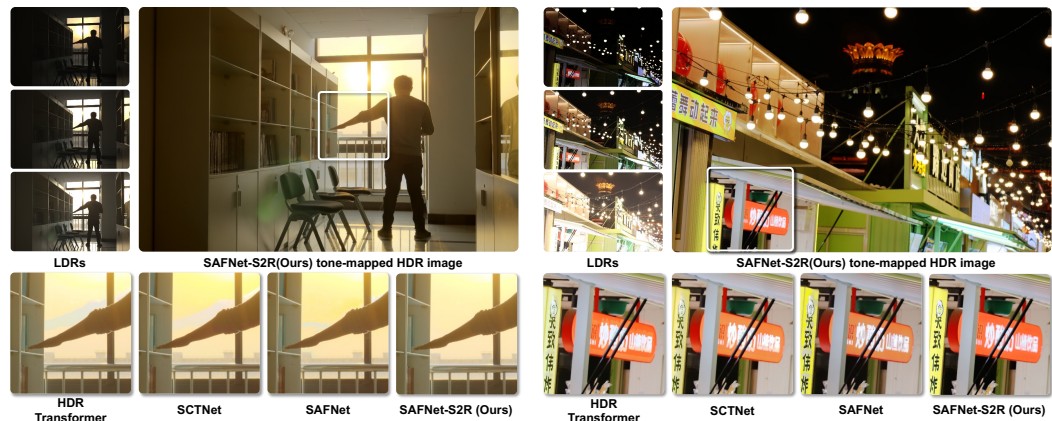

Figure 7: Visual results on real-captured scenes show our solution reduces ghosting in backlit scenes (left) and recovers highlights (right).

Table 4: Experimental results on the effectiveness of the S2R-HDR dataset. We test the cross-dataset generalization of models trained on different datasets. Models trained on our S2R-HDR dataset achieve superior generalization and require only minimal fine-tuning on SCT or Challenge123 to reach state-of-the-art performance. The best results are in **bold**.

| Methods | Training | Testing on SCT (Tel et al., 2023) | | | | Testing on Challenge123 (Kong et al., 2024) | | | |
|---|---|---|---|---|---|---|---|---|---|
| | | PSNR-$\mu$ | PSNR-$\ell$ | SSIM-$\mu$ | SSIM-$\ell$ | PSNR-$\mu$ | PSNR-$\ell$ | SSIM-$\mu$ | SSIM-$\ell$ |
| HDR-Transformer (Liu et al., 2022) | SCT (Tel et al., 2023) | 42.39 | 46.35 | 0.9844 | 0.9948 | 34.48 | 24.60 | 0.9744 | 0.9573 |
| | Challenge123 (Kong et al., 2024) | 31.94 | 34.23 | 0.9518 | 0.9503 | 40.70 | 28.72 | 0.9881 | 0.9731 |
| | SCT & Challenge123 | 42.09 | 44.64 | 0.9864 | 0.9947 | 39.52 | 28.23 | 0.9892 | 0.9744 |
| | S2R-HDR | 34.89 | 41.67 | 0.9575 | 0.9926 | 41.51 | 30.06 | 0.9870 | 0.9787 |
| | S2R-HDR Fine-tune on SCT or Challenge123 | **43.25** | **47.36** | **0.9877** | **0.9957** | **42.40** | **30.48** | **0.9912** | **0.9797** |
| SCTNet (Tel et al., 2023) | SCT (Tel et al., 2023) | 42.55 | **47.51** | 0.9850 | 0.9952 | 34.57 | 25.07 | 0.9753 | 0.9599 |
| | Challenge123 (Kong et al., 2024) | 32.60 | 35.93 | 0.9535 | 0.9639 | 40.65 | 28.73 | 0.9882 | 0.9721 |
| | SCT & Challenge123 | 40.00 | 42.79 | 0.9800 | 0.9935 | 40.04 | 28.21 | 0.9898 | 0.9750 |
| | S2R-HDR | 34.83 | 42.32 | 0.9526 | 0.9933 | 41.49 | **30.37** | 0.9862 | 0.9796 |
| | S2R-HDR Fine-tune on SCT or Challenge123 | **43.22** | 47.28 | **0.9872** | **0.9961** | **42.10** | 30.18 | **0.9914** | **0.9798** |
| SAFNet (Kong et al., 2024) | SCT (Tel et al., 2023) | 42.66 | 48.38 | 0.9831 | 0.9955 | 34.26 | 25.50 | 0.9718 | 0.9590 |
| | Challenge123 (Kong et al., 2024) | 35.14 | 38.77 | 0.9619 | 0.9868 | 41.88 | 29.73 | 0.9897 | 0.9784 |
| | SCT & Challenge123 | 42.12 | 45.14 | 0.9853 | 0.9941 | 41.61 | 29.72 | 0.9901 | 0.9788 |
| | S2R-HDR | 34.89 | 43.85 | 0.9500 | 0.9939 | 42.75 | **32.11** | 0.9872 | 0.9822 |
| | S2R-HDR Fine-tune on SCT or Challenge123 | **43.03** | **48.79** | **0.9831** | **0.9958** | **43.30** | 31.59 | **0.9914** | **0.9819** |

We train the three baseline methods on the SCT dataset, the Challenge123 dataset, a merged dataset combining SCT and Challenge123, and our S2R-HDR dataset, then evaluate their generalization by testing on both SCT and Challenge123. Furthermore, given the domain gap between synthetic datasets (such as S2R-HDR) and real-world datasets (SCT and Challenge123), we also fine-tune the models trained on the synthetic S2R-HDR dataset on the real datasets, following the approach in Niklaus et al. (2021).

As shown in Table 4, the model trained on our dataset surprisingly outperforms the one trained directly on Challenge123 when evaluated on the same dataset. Moreover, models trained on either the SCT or Challenge123 datasets suffer significant performance degradation during cross-validation, indicating their limited generalization capability. In contrast, models trained solely on our S2R-HDR dataset—without any exposure to SCT or Challenge123—demonstrate superior cross-dataset generalization, highlighting the high quality and robustness of our dataset. Additionally, models trained on S2R-HDR require only minimal fine-tuning on SCT or Challenge123 to achieve state-of-the-art performance. Across all three tested methods, models trained on S2R-HDR outperformed those trained directly on SCT or Challenge123, achieving at least a 0.4 dB improvement in PSNR-$\mu$. These results confirm the effectiveness of our S2R-HDR dataset in enhancing model robustness and generalization for HDR fusion tasks. We further show the visualization results of our S2R-HDR dataset comparison experiments in Appendix A.5. We also compare our S2R-HDR dataset with the Kalantari (Kalantari & Ramamoorthi, 2017) and Real-HDRV(Deghosting) (Shu et al., 2024) datasets in Appendix A.6.

**Effectiveness of S2R-Adapter's two branches.** To validate the effectiveness of the knowledge-sharing branch and knowledge-transfer branch designed in our Adapter method, we conducted ab-

Table 5: Ablation study of the S2R-Adapter using the SAFNet model (Kong et al., 2024) pre-trained on the S2R-HDR dataset. The experiments are conducted on the SCT dataset (Tel et al., 2023). When both branches work together with learned scale factors $\alpha_s$ and $\alpha_t$, optimal performance is achieved. In the case of non-learnable $\alpha_s$ and $\alpha_t$, their values are set to 1.

| Baseline | Fine-tune | Share | Transfer | Learned | PSNR-$\mu$ | PSNR-$\ell$ | SSIM-$\mu$ | SSIM-$\ell$ |
|---|---|---|---|---|---|---|---|---|
| ✓ | | | | | 34.89 | 43.85 | 0.9500 | 0.9939 |
| | ✓ | | | | 43.03 | 48.79 | 0.9831 | 0.9958 |
| | | ✓ | | | 43.32 | 48.76 | 0.9860 | 0.9958 |
| | | | ✓ | | 43.20 | 47.61 | 0.9855 | 0.9957 |
| | | ✓ | ✓ | | 43.28 | 48.68 | 0.9863 | 0.9959 |
| | | ✓ | ✓ | ✓ | 43.33 | 48.90 | 0.9864 | 0.9959 |

Table 6: Ablation study of the S2R-Adapter Framework under test-time adaptation without GT data. The baseline is the SAFNet (Kong et al., 2024) pre-trained on S2R-HDR Dataset. The test data is the SCT Dataset. TS stands for the teacher-student framework, Adapter for shared and target branch adapters, and Unc for scale factor adjustment with uncertainty.

| Baseline | TS | Adapter | Unc | PSNR-$\mu$ | PSNR-$\ell$ | SSIM-$\mu$ | SSIM-$\ell$ |
|---|---|---|---|---|---|---|---|
| ✓ | | | | 34.89 | 43.85 | 0.9500 | 0.9939 |
| ✓ | ✓ | | | 34.93 | 44.71 | 0.9477 | 0.9944 |
| ✓ | ✓ | ✓ | | 36.05 | 46.79 | 0.9469 | 0.9948 |
| ✓ | ✓ | ✓ | ✓ | 36.28 | 47.23 | 0.9586 | 0.9949 |

Table 7: Experiment results on knowledge control using SAFNet (Kong et al., 2024) on the SCT (Tel et al., 2023) and S2R-HDR datasets. The result shows that our S2R-Adapter better alleviates knowledge forgetting.

| SAFNet (Kong et al., 2024) | Test on SCT (Tel et al., 2023) | | Test on S2R-HDR | |
|---|---|---|---|---|
| | PSNR-$\mu$ | PSNR-$\ell$ | PSNR-$\mu$ | PSNR-$\ell$ |
| Fine-tune on SCT (Tel et al., 2023) | 43.03 | 48.79 | 35.52 | 29.40 |
| S2R-Adapter on SCT (Tel et al., 2023) | **43.33** | **48.90** | **35.95** | **29.80** |

lation experiments on the SCT dataset using SAFNet as the baseline to evaluate the impact of each branch on the experimental results. As shown in Table 5, we tested the effect of using each branch individually. Results indicate that using only the knowledge-sharing branch outperforms simple fine-tuning, suggesting that this branch effectively learns shared knowledge, thereby reducing the forgetting of pre-trained knowledge. Meanwhile, using only the knowledge-transfer branch leads to a more substantial improvement, further confirming the significant differences between synthetic and real data. When both branches work together with learned scale factors $\alpha_s$ and $\alpha_t$, optimal performance is achieved.

**Effectiveness of knowledge control.** We conduct experiments to show that our method better facilitates knowledge control than simple fine-tuning, effectively alleviating knowledge forgetting. Specifically, we first train the SAFNet (Kong et al., 2024) on the S2R-HDR as a pre-trained model. Then, we apply simple fine-tuning and our adapter-based fine-tuning for domain adaptation on the SCT (Tel et al., 2023) dataset and subsequently test the models on the original S2R-HDR training set to measure knowledge forgetting. As shown in Table 7, S2R-Adapter effectively adapts to the SCT dataset while minimizing knowledge forgetting, demonstrating better preservation of pre-trained knowledge.

**Effectiveness of S2R-Adapter framework under test-time adaptation.** We validate the S2R-Adapter Framework's effectiveness during test-time adaptation through ablation studies on the SCT dataset, using SAFNet as the baseline, pre-trained on our S2R-HDR dataset. As shown in Table 6, the teacher-student framework enhances results by making the test-time adaptation process more robust. Most improvements are from our shared and transfer branch adapters. Additionally, dynamically adjusting the scale factor between the adapter branches based on uncertainty measurement allows for better control of shared and transferred knowledge across varying domain shifts, further enhancing performance.

## 6 CONCLUSION

This paper introduces the S2R-HDR dataset, a large-scale, high-quality resource for HDR fusion in dynamic scenes. By providing diverse, controllable, and high-fidelity synthetic data, the dataset addresses the limitations of existing HDR datasets. Additionally, we propose the S2R-Adapter, a novel domain adaptation method that effectively bridges the gap between synthetic and real data, enabling efficient knowledge transfer. Experimental results on both labeled and unlabeled datasets demonstrate that our S2R-HDR dataset and S2R-Adapter significantly enhance the performance of HDR fusion models in real-world scenarios. This provides a viable solution for the HDR field, where data acquisition is often limited. Future work will focus on expanding the S2R-HDR dataset to support a wider range of application scenarios.

## 7 REPRODUCIBILITY STATEMENT

To ensure the reproducibility of our findings, detailed implementation instructions are provided in Section 5 and Appendix A.1. In addition, the source code and datasets are publicly available at URL: https://openimaginglab.github.io/S2R-HDR/.

## 8 ACKNOWLEDGEMENT

This work is supported/funded in part by the Shanghai Artificial Intelligence Laboratory, the Centre for Perceptual and Interactive Intelligence (CPII) Ltd., a CUHK-led InnoCentre under the InnoHK initiative of the Innovation and Technology Commission of the Hong Kong Special Administrative Region Government, RGC Early Career Scheme (ECS) No. 24209224, CUHK-CUHK(SZ)-GDSTC Joint Collaboration Fund No. 2025A0505000053. We thank Yixuan Li for helpful discussions on Unreal Engine rendering. We also thank Haoran Yang for his assistance in capturing the real-world dataset.

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

## A  ADDITIONAL EXPERIMENTS

### A.1  EXPERIMENTS DETAILS

When training these methods on our S2R-HDR dataset, we first generate three different exposure ({-2, 0, +2)}, {-3, 0, +3}) LDR images from the original HDR images. Following this, we apply the same data augmentation techniques, and training schedules across all models. Additionally, we introduce random Gaussian noise with $\sigma \in [0.0001, 0.001]$ to the lowest exposure image and $\sigma \in [0.00001, 0.0001]$ to the middle exposure image.

For the SCTNet (Tel et al., 2023) architecture, which is based on the Transformer framework, we employ a linear layer as the projection layer of the S2R-Adapter (as illustrated in the left part of Figure 4) and integrate it into SCTNet's *WindowAttention Linear Layer*. In contrast, for the SAFNet (Kong et al., 2024) architecture, which is based on CNNs, we utilize a $1\times1$ convolutional layer as the S2R-Adapter's projection layer and inject it into the network at layers indexed by *[3:25:2]* and *[42:58:4]*. For both CNN and Transformer architecture, we set the rank of the shared branch adapter $r_s$ to be 1, and the rank of the transfer branch $r_t$ to be 64, following Liu et al. (2023a).

In our test-time adaptation experiments, each test sample is processed only once. To assess sample uncertainty as a measure of domain shift, we employ test-time augmentation techniques. Specifically, we augment test samples using a variety of exposure levels: $[-0.1, -0.5, 0, 0.5, 1]$. Additionally, we apply random transformations, including flips, white balance adjustments, and random Gaussian noise. For augmentations involving exposure and white balance, we apply the parameters to the input images following inverse tone mapping. Correspondingly, the inverse transformations are directly applied to the model outputs. This results in $N$ augmented outputs per sample, from which we compute the variance across these outputs to quantify uncertainty $\mathcal{U}(x)$.

We used the Photomatix software to perform tone mapping on HDR images.

### A.2  ANALYSIS OF DOMAIN GAP AND ADAPTER

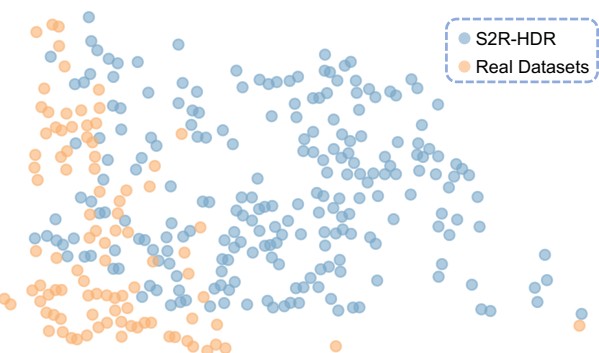

Figure 8: The distribution of our S2R-HDR dataset and real captured HDR datasets (Kalantari & Ramamoorthi, 2017; Tel et al., 2023; Kong et al., 2024). Following the approach outlined in Shu et al. (2024); Guo et al. (2023); Hu et al. (2022b), for each HDR image, we first extract 7-dimensional features listed in Table 18 that capture key aspects of HDR, including the extent of dynamic range, intra-frame diversity, and the overall style of the HDR images. These features are then projected into a 2D space using t-SNE (Van der Maaten & Hinton, 2008) for visualization. Regarding the concern about real-dataset sample size, we analyzed all real images (Kalantari & Ramamoorthi, 2017; Tel et al., 2023; Kong et al., 2024) without sampling bias.

**Distribution of our S2R-HDR dataset vs. real datasets.** To better understand the distribution of our S2R-HDR dataset compared to existing real-world HDR datasets (Kalantari & Ramamoorthi, 2017; Tel et al., 2023; Kong et al., 2024), we extract seven-dimensional features representing key HDR characteristics, including dynamic range, intra-frame diversity, and overall image style. These features are visualized using t-SNE (Van der Maaten & Hinton, 2008), as illustrated in Figure 8. The

results demonstrate that our S2R-HDR dataset spans a significantly broader range of HDR scenarios, reflecting richer variations in lighting conditions, scene dynamics, and environmental styles.

**Why we need an adapter.** Despite the broader coverage of S2R-HDR, there remains a noticeable domain gap between our synthetic data and real HDR datasets, as highlighted in Figure 8. This gap primarily stems from differences in texture representation, environmental details, and natural lighting conditions. To address this, we introduce an S2R-Adapter to bridge the disparity, enabling models trained on synthetic data to generalize effectively to real-world scenes.

**What the adapter learns.** To gain deeper insights into the domain gap between real and rendered data and to better understand what domain adaptation learns, we compute difference maps for models trained on the rendered dataset (S2R-HDR) before and after domain adaptation (S2R-Adapter) to the SCT dataset. As shown in Figure 9, the differences are primarily concentrated in regions containing trees, grass, and people, while ground, sky, and buildings remain largely unchanged. This suggests that the key discrepancies between real and rendered data mainly arise in texture-rich areas such as human figures and vegetation. The results further confirm that domain adaptation effectively mitigates the domain gap.

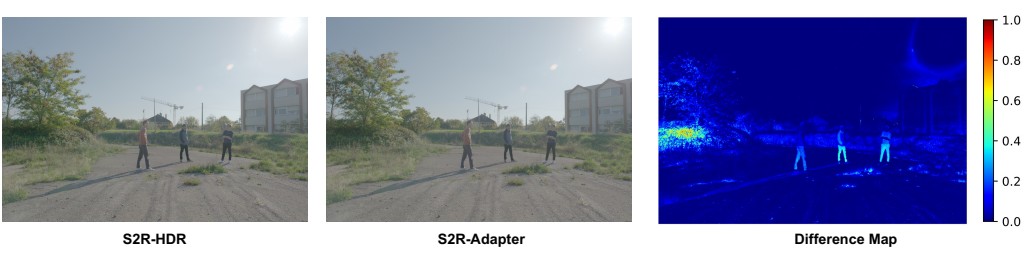

Figure 9: Difference maps of models trained on the rendered dataset (S2R-HDR) before and after domain adaptation (S2R-Adapter) to the SCT dataset. The differences are primarily concentrated in texture-rich regions such as trees, grass, and people, while ground, sky, and buildings remain largely unchanged. This highlights that the key domain discrepancies lie in fine textures and demonstrates the effectiveness of domain adaptation in bridging the domain gap.

### A.3 Domain Adaptation Method Comparison (Fine-tune VS S2R-Adapter)

As emphasized, the key challenge lies in addressing the domain gap. We summarize the domain adaptation experiments using fine-tuning and the S2R-Adapter. As shown in Table 8, regardless of the method employed, mitigating the domain gap consistently leads to notable performance improvements.

As shown in Table 8, the performance of SCTNet and SAFNet was evaluated on the SCT and Challenge123 datasets after being trained on different datasets. Since SCTNet is specifically designed for the SCT dataset, it already performs well on this dataset. Consequently, using the S2R-Adapter results in only a modest improvement compared to fine-tuning SCTNet on the SCT dataset. However, on the Challenge123 dataset, the use of the S2R-Adapter leads to a more substantial improvement over fine-tuning SCTNet alone, with PSNR-$\mu$ increasing from 42.1 dB to 42.58 dB. Similarly, SAFNet, which is designed for the Challenge123 dataset, shows a more significant performance improvement on the SCT dataset, with PSNR-$\mu$ increasing from 43.03 dB to 43.33 dB. The key challenge remains addressing the domain gap, and mitigating this gap consistently results in significant performance gains.

We further conduct comparative experiments to evaluate the effectiveness of our dataset against existing datasets in an adapter-based setting. Specifically, we first pre-train the SAFNet (Kong et al., 2024) on the S2R-HDR and Challenge123 (Kong et al., 2024) datasets, respectively, and then adapt them to the SCT dataset using the S2R-Adapter. As shown in Table 9, models pretrained on the S2R-HDR dataset achieve better performance after adaptation. Moreover, we observe that using the S2R-Adapter consistently outperforms both training solely on the SCT dataset and joint training on SCT and Challenge123 datasets, demonstrating the advantage of the proposed adaptation strategy.

Moreover, as shown in Table 7 in the main text, our S2R-Adapter not only improves performance but also offers better knowledge control, with reduced knowledge forgetting compared to fine-tuning. In addition to fine-tuning on labeled real data, our S2R-Adapter demonstrates superior performance in test-time adaptation, allowing flexible adaptation to unseen data without the need for ground-truth labels. This is particularly beneficial for HDR fusion tasks, where large-scale, accurately labeled real-world datasets are often unavailable. By pretraining on large synthetic datasets and adapting to any unlabeled target distribution during test time, our approach exhibits greater flexibility and generalization.

Table 8: Experimental results on the effectiveness of the S2R-HDR dataset. We test the cross-dataset generalization of models trained on different datasets. Models trained on our S2R-HDR dataset achieve superior generalization and require only minimal fine-tuning on SCT or Challenge123 to reach state-of-the-art performance. The best results are in **bold**.

| Methods | Training | Testing on SCT (Tel et al., 2023) | | | | Testing on Challenge123 (Kong et al., 2024) | | | |
|---|---|---|---|---|---|---|---|---|---|
| | | PSNR-$\mu$ | PSNR-$\ell$ | SSIM-$\mu$ | SSIM-$\ell$ | PSNR-$\mu$ | PSNR-$\ell$ | SSIM-$\mu$ | SSIM-$\ell$ |
| SCTNet (Tel et al., 2023) | SCT (Tel et al., 2023) | 42.55 | 47.51 | 0.9850 | 0.9952 | 34.57 | 25.07 | 0.9753 | 0.9599 |
| | Challenge123 (Kong et al., 2024) | 32.60 | 35.93 | 0.9535 | 0.9639 | 40.65 | 28.73 | 0.9882 | 0.9721 |
| | SCT & Challenge123 | 40.00 | 42.79 | 0.9800 | 0.9935 | 40.04 | 28.21 | 0.9898 | 0.9750 |
| | S2R-HDR | 34.83 | 42.32 | 0.9526 | 0.9933 | 41.49 | 30.37 | 0.9862 | 0.9796 |
| | S2R-HDR Fine-tune on SCT and Challenge123 | 41.40 | 46.37 | 0.9820 | 0.9960 | 41.93 | 30.33 | 0.9907 | 0.9796 |
| | S2R-HDR Fine-tune on SCT or Challenge123 | 43.22 | 47.28 | **0.9872** | 0.9961 | 42.10 | 30.18 | 0.9914 | 0.9798 |
| | S2R-HDR w S2R-Adapter | **43.24** | **48.32** | **0.9872** | **0.9962** | **42.58** | **30.68** | **0.9915** | **0.9805** |
| SAFNet (Kong et al., 2024) | SCT (Tel et al., 2023) | 42.66 | 48.38 | 0.9831 | 0.9955 | 34.26 | 25.50 | 0.9718 | 0.9590 |
| | Challenge123 (Kong et al., 2024) | 35.14 | 38.77 | 0.9619 | 0.9868 | 41.88 | 29.73 | 0.9897 | 0.9784 |
| | SCT & Challenge123 | 42.12 | 45.14 | 0.9853 | 0.9941 | 41.61 | 29.72 | 0.9901 | 0.9788 |
| | S2R-HDR | 34.89 | 43.85 | 0.9500 | 0.9939 | 42.75 | 32.11 | 0.9872 | 0.9822 |
| | S2R-HDR Fine-tune on SCT and Challenge123 | 42.97 | 47.84 | 0.9861 | **0.9960** | 42.91 | 31.05 | 0.9906 | 0.9805 |
| | S2R-HDR Fine-tune on SCT or Challenge123 | 43.03 | 48.79 | 0.9831 | 0.9958 | 43.30 | 31.59 | 0.9914 | 0.9819 |
| | S2R-HDR w S2R-Adapter | **43.33** | **48.90** | **0.9864** | 0.9959 | **43.43** | **31.84** | **0.9915** | **0.9824** |

Table 9: Effectiveness of S2R-HDR and S2R-Adapter. Pretraining SAFNet (Kong et al., 2024) on S2R-HDR and adapting to SCT (Tel et al., 2023) yields the best overall results.

| Training | PNSR-$\mu$ | PNSR-$\ell$ | SSIM-$\mu$ | SSIM-$\ell$ |
|---|---|---|---|---|
| SCT (Tel et al., 2023) | 42.66 | 48.38 | 0.9831 | 0.9955 |
| Challeng123 (Kong et al., 2024) | 35.14 | 38.77 | 0.9619 | 0.9868 |
| SCT & Challenge | 42.12 | 45.14 | 0.9853 | 0.9941 |
| S2R-HDR with S2R-Adapter on SCT | **43.44** | **48.90** | **0.9864** | **0.9959** |
| Challenge123 with S2R-Adapter on SCT | 42.92 | 47.05 | 0.9856 | 0.9951 |

## A.4 ADDITIONAL RESULTS ON REAL-CAPTURE IMAGES

We further provide a visual comparison using real-captured data without ground truth in Figure 10. Our approach effectively reduces artifacts in challenging scenarios.

## A.5 VISUALIZATION OF EFFECTIVENESS OF S2R-HDR

We further show the visualization results of our S2R-HDR dataset comparison experiments in Figure 11, models trained on our S2R-HDR dataset achieve optimal visual quality compared to those trained on other datasets. Additionally, as depicted in the left image of Figure 11, our dataset effectively mitigates motion occlusion challenges. Similarly, as shown in the right image of Figure 11, our dataset effectively addresses challenges related to high light fusion.

## A.6 ADDITIONAL DATA EFFECTIVENESS COMPARISON EXPERIMENTS

To validate the effectiveness of our S2R-HDR dataset, we used SCTNet (Tel et al., 2023) as the baseline model and conducted experiments on the Real-HDRV (Deghosting) (Shu et al., 2024) dataset, which, although the largest, is less commonly used. The results, as shown in Table 10, with simple fine-tuning, our dataset consistently delivers the best results.

We also use SCTNet (Tel et al., 2023) as a baseline model and conduct experiments on the earliest Kalantari (Kalantari & Ramamoorthi, 2017) dataset. SCTNet is retrained on the entire dataset, and

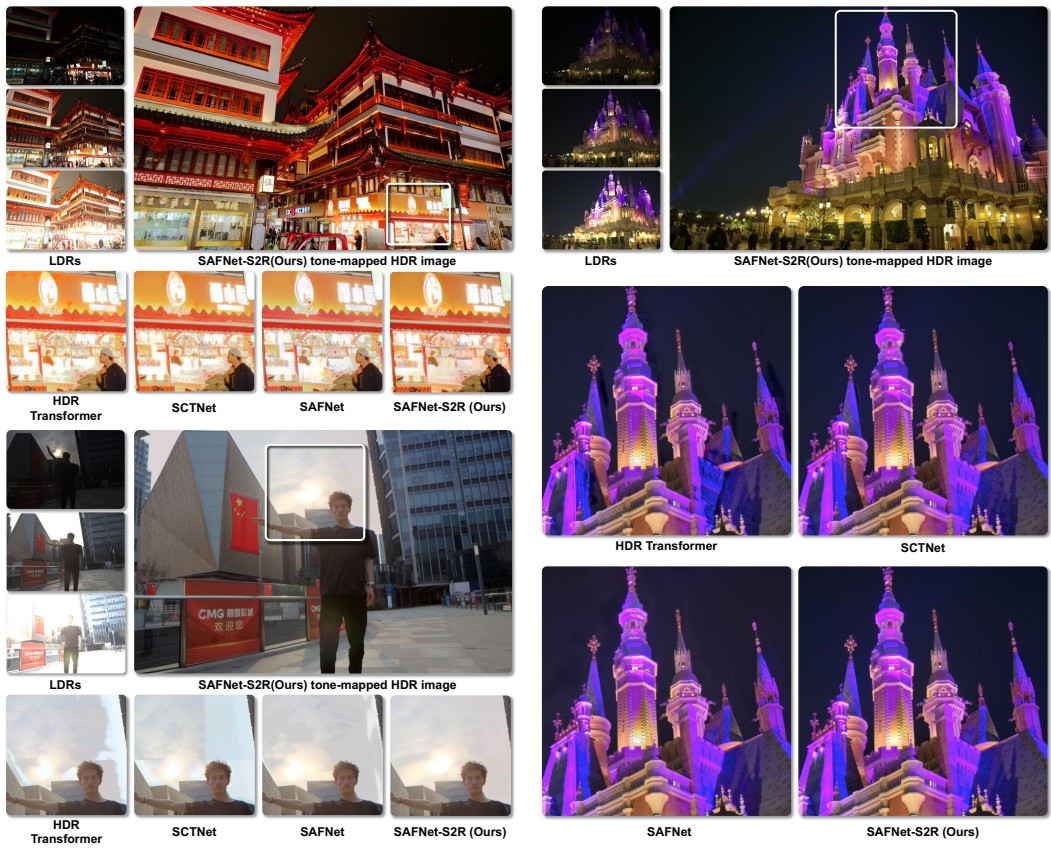

Figure 10: Visualization results on real-captured data without ground truth. Our approach effectively reduces artifacts in highlight areas and alleviates ghosting in nighttime scenarios.

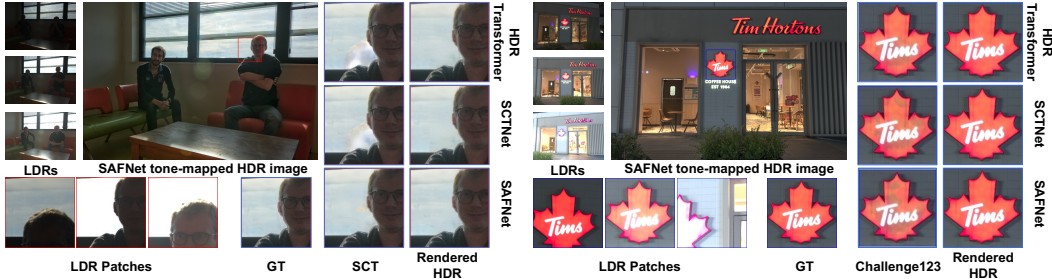

Figure 11: Visualization results of our S2R-HDR dataset comparison experiments. Models trained on our S2R-HDR dataset exhibit significantly fewer artifacts compared to those trained on the SCT dataset (Tel et al., 2023) or Challenge123 dataset (Kong et al., 2024).

the results, as shown in Table 11, demonstrate that with simple fine-tuning, our dataset consistently yields the best performance.

## A.7  ADDITIONAL EXPERIMENTS ON THE EFFECTIVENESS OF S2R-ADAPTER

We conducted additional experiments on more datasets to further evaluate the effectiveness of S2R-Adapter. These experiments include test-time adaptation without ground truth on Kalantari's (Kalantari & Ramamoorthi, 2017), Sen's (Sen et al., 2012), and Tursun's (Tursun et al., 2016) datasets.

Table 10: Experimental results of data effectiveness comparison with the Real-HDRV (Deghosting) (Shu et al., 2024) Datasets.

| SCTNet | SCT | | | | Challenge123 | | | |
|---|---|---|---|---|---|---|---|---|
| | PSNR-$\mu$ | PSNR-$\ell$ | SSIM-$\mu$ | SSIM-$\ell$ | PSNR-$\mu$ | PSNR-$\ell$ | SSIM-$\mu$ | SSIM-$\ell$ |
| Train on Real-HDRV (Deghosting) | 35.37 | 46.13 | 0.9651 | 0.9949 | 36.41 | 26.42 | 0.9711 | 0.9674 |
| Fine-tune on SCT/Challenge123 | 42.98 | 47.27 | **0.9880** | 0.9956 | 40.84 | 28.91 | 0.9905 | 0.9765 |
| Train on S2R-HDR | 34.83 | 42.32 | 0.9526 | 0.9933 | 41.49 | **30.37** | 0.9862 | 0.9796 |
| Fine-tune on SCT/Challenge123 | **43.22** | **47.28** | 0.9872 | **0.9961** | **42.10** | 30.18 | **0.9914** | **0.9798** |

Table 11: Experimental results of data effectiveness comparison on the earliest Kalantari (Kalantari & Ramamoorthi, 2017) Datasets.

| Methods | Training | PSNR-$\mu$ | PSNR-$\ell$ | SSIM-$\mu$ | SSIM-$\ell$ |
|---|---|---|---|---|---|
| | Kalantari | 44.13 | 42.32 | 0.9916 | 0.9890 |
| SCTNet | S2R-HDR | 41.41 | 36.68 | 0.9859 | 0.9787 |
| | S2R-HDR Fine-tune on Kalantari | **44.32** | **43.16** | **0.9923** | **0.9911** |

We first performed test-time adaptation without ground-truth labels on Kalantari's dataset using the SAFNet. Table 12 presents the results.

Table 12: Test-time adaptation results on Kalantari (Kalantari & Ramamoorthi, 2017) dataset.

| Methods | PSNR-$\mu$ | PSNR-$\ell$ | SSIM-$\mu$ | SSIM-$\ell$ |
|---|---|---|---|---|
| SAFNet trained on SCT | 34.75 | 39.41 | 0.9806 | 0.9821 |
| SAFNet trained on Challenge123 | 40.31 | 37.19 | 0.9843 | 0.9772 |
| SAFNet trained on S2R-HDR | 43.06 | 40.63 | 0.9887 | 0.9878 |
| SAFNet trained on S2R-HDR with S2R-Adapter | **43.29** | **41.488** | **0.9889** | **0.9890** |

We also conducted test-time adaptation on Sen's (Sen et al., 2012) and Tursun's (Tursun et al., 2016) datasets. We report the non-reference IQA metric MUSIQ (Ke et al., 2021) scores in Table 13.

Table 13: Test-time adaptation results (MUSIQ scores) on Sen's (Sen et al., 2012) and Tursun's (Tursun et al., 2016) datasets.

| Methods | MUSIQ on Tursun's | MUSIQ on Sen's |
|---|---|---|
| SAFNet trained on SCT | 63.94 | 65.71 |
| SAFNet trained on Challenge123 | 63.42 | 65.65 |
| SAFNet trained on S2R-HDR | 63.32 | 65.97 |
| SAFNet trained on S2R-HDR with S2R-Adapter | **65.30** | **67.45** |

## A.8 ADDITIONAL RESULTS ON HDR VIDEO RECONSTRUCTION TASK

We also conduct experiments on HDR video reconstruction task to validate the effectiveness of our S2R-HDR dataset. We use recent open-sourced methods, HDRFlow (Xu et al., 2024), as our baseline model, and conduct training experiments on Vimeo-90K (Xue et al., 2019), Sintel (Butler et al., 2012), and Real-HDRV (Shu et al., 2024) datasets, testing on DeepHDRVideo (Chen et al., 2021) dataset. The results, as shown in Table 14, with simple fine-tuning, our dataset consistently delivers the best results.

Additionally, in Table 15, we tested temporal consistency using the TOG HDR dynamic dataset (Kalantari et al., 2013) with method (Lai et al., 2018), this metric is calculated using Equation 4, which measures the temporal consistency of a video by quantifying the flow warping error between adjacent frames. In this equation, $V_t$ represents the original frame, $\hat{V}_{t+1}$ represents the warped frame using optical flow and $M$ is a non-occlusion mask indicating non-occluded regions. The results show that our dataset achieves the best performance in temporal consistency.

Table 14: Experimental results of data effectiveness comparison on the HDR video reconstruction task. The HDRFlow method is trained on different datasets and tested on the DeepHDRVideo (Chen et al., 2021) dataset.

| Methods | Training | PSNR-$\mu$ | PSNR-$\ell$ | SSIM-$\mu$ | SSIM-$\ell$ |
|---|---|---|---|---|---|
| | Vimeo+Sintel | 43.25 | 53.05 | 0.9520 | 0.9956 |
| HDRFlow (Xu et al., 2024) | Real-HDRV | 43.37 | 53.05 | 0.9540 | 0.9958 |
| | S2R-HDR | 43.13 | 52.50 | 0.9508 | 0.9953 |
| | S2R-HDR Fine-tune on Real-HDRV | **43.51** | **53.52** | **0.9546** | **0.9960** |

Table 15: Experimental results of temporal consistency comparison on the HDR video reconstruction task.

| Training Datasets | Sintel+Vimeo | HDRV | S2R-HDR |
|---|---|---|---|
| Temporal Stability Score↓ | 0.2201 | 0.2183 | 0.1773 |

$$E_{\text{warp}}\left(V_t, V_{t+1}\right) = \frac{1}{\sum_{i=1}^{N} M_t^{(i)}} \sum_{i=1}^{N} M_t^{(i)} \left\| V_t^{(i)} - \hat{V}_{t+1}^{(i)} \right\|_2^2$$
$$E_{\text{warp}}(V) = \frac{1}{T-1} \sum_{t=1}^{T-1} E_{\text{warp}}\left(V_t, V_{t+1}\right) \tag{4}$$

## A.9 DETAILED DATASET GENERATION PIPELINE

We provide a complete and transparent overview of our dataset-generation pipeline in Figure12. Because this pipeline allows us to directly control the lighting parameters of the sources in the UE5 simulation environment (including their positions, intensities, etc.), we can synthesize a highly diverse set of illumination conditions. In addition, we can manipulate the pose and motion of objects in the scene by binding different animation sequences to them, thereby producing a wide variety of motion patterns. With the exception of the use of several high-quality UE5 scene assets that we purchased for scene construction, the rest of the pipeline is fully reproducible.

## A.10 DETAILED DATASET SCALE COMPARISON

We include a comparison of dataset scales between our S2R-HDR dataset and the datasets from SCT (Tel et al., 2023), Challenge123 (Kong et al., 2024), Kalantari (Kalantari & Ramamoorthi, 2017), and Real-HDRV(Deghosting) (Shu et al., 2024), as presented in Table 16.

Table 16: Comparison of dataset scale. We compare our S2R-HDR dataset with SCT (Tel et al., 2023), Challenge123 (Kong et al., 2024), Kalantari (Kalantari & Ramamoorthi, 2017), and Real-HDRV(Deghosting) (Shu et al., 2024).

| | SCT (Tel et al., 2023) | Challenge123 (Kong et al., 2024) | Kalantari (Kalantari & Ramamoorthi, 2017) | Real-HDRV(Deghosting) (Shu et al., 2024) | S2R-HDR |
|---|---|---|---|---|---|
| Dataset size | 144 (108/36) | 123 (96/27) | 89 (74/15) | 500 (450/50) | 24,000 |

## A.11 FURTHER SCENE-LEVEL ANALYSIS OF S2R-HDR

We provide additional scene-level analysis of our proposed dataset S2R-HDR. First, following the same methodology as in (Shu et al., 2024), we analyze the directions of motion using the optical flow. Figure13 reports the proportion of pixels corresponding to different motion directions in S2R-HDR, from which it can be observed that the motion patterns in our dataset are highly diverse. Second, Table19 summarizes the distribution of illumination conditions and locations. In terms of scene semantics, our dataset covers a wide variety of environments, including home interiors, urban streets, churches, rural areas, parks, subway stations, workshops, and more. Regarding the foreground assets, Figure14 further illustrates the richness of our foreground content together with the associated motion patterns.

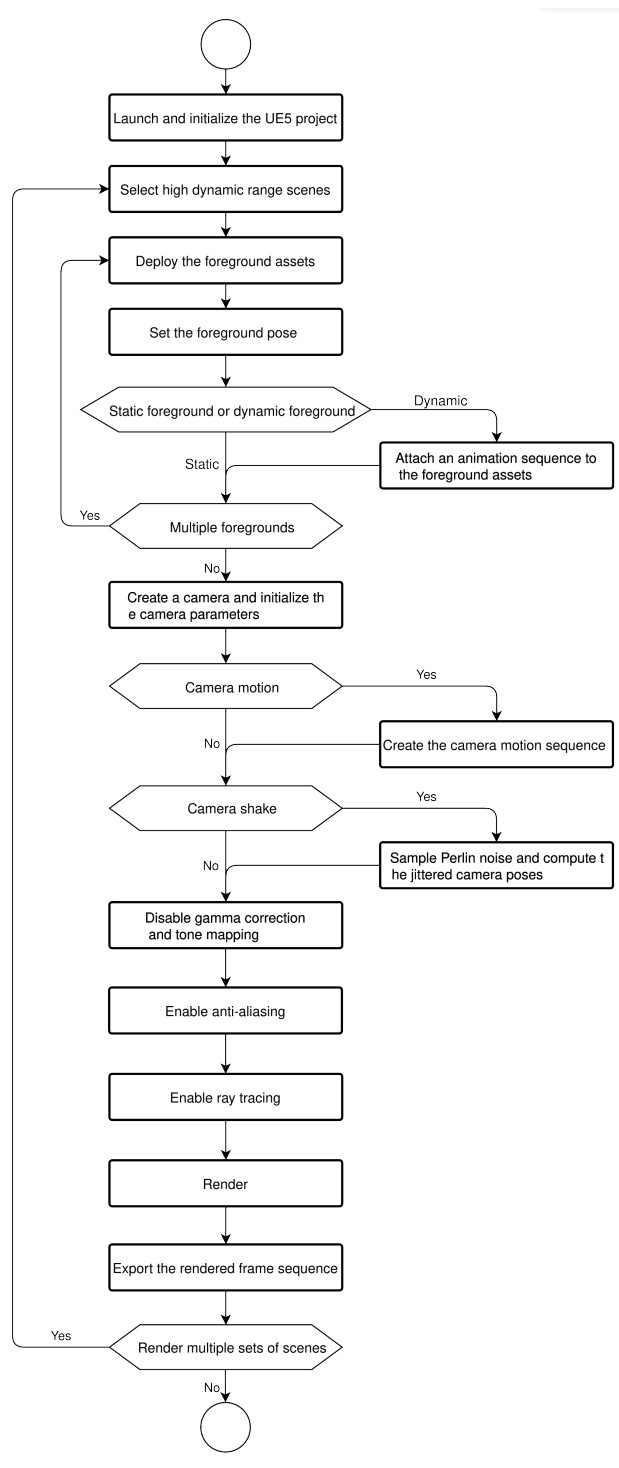

Figure 12: Distribution of optical flow directions across S2R-HDR.

## A.12  TEST-TIME ADAPTATION RUNTIME

Like most test-time adaptation (TTA) methods (Kundu et al., 2020; Liu et al., 2023a; Wang et al., 2022; Sun et al., 2020; Wang et al., 2020), our approach incurs additional computational cost due

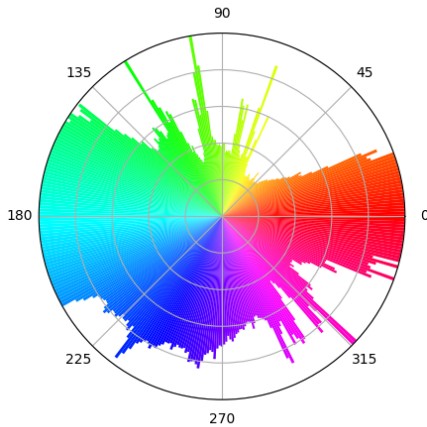

Figure 13: Distribution of optical flow directions across S2R-HDR.

to model weight updates during inference to adapt to the target domain without ground-truth labels. However, this cost is modest and justified by the performance gains.

To quantify the overhead, we evaluated test-time adaptation runtime on SCT test images (resolution 1500×1000) using SAFNet as the base model. Experiments were conducted on a server with an AMD EPYC 7402 (48C) @ 2.8 GHz CPU, 8×NVIDIA RTX 4090 GPUs, 512 GB RAM, running CentOS 7.9. The results are as follows:

Table 17: Test-time Adaptation Runtime

|  | Standard Testing | Test-Time Adaptation |
|---|---|---|
| Runtime (seconds) | 0.3813 | 0.6128 |
| PSNR-tested on SCT dataset | 43.85 | 47.23 |

The TTA process introduces an additional 0.23 seconds per image, with a performance improvement of +3.38 PSNR. This trade-off is typical for TTA methods and demonstrates effective adaptation to unseen data. The overhead arises from the model updating its weights to better align with the target domain distribution, without requiring ground-truth labels. This extra computation is common in test-time adaptation methods (Kundu et al., 2020; Liu et al., 2023a; Wang et al., 2022; Sun et al., 2020; Wang et al., 2020), as they rely on on-the-fly optimization to adapt to distribution shifts during inference.

Table 18: Metrics to assess the diversity of different HDR datasets.

| FHLP | Fraction of HighLight Pixel (Guo et al., 2023) |
|---|---|
| EHL | Extent of HighLight (Guo et al., 2023) |
| SI | Spatial Information (Series, 2012) |
| CF | ColorFulness (Hasler & Suesstrunk, 2003) |
| stdL | standard deviation of Luminance (Guo et al., 2023) |
| ALL | Average Luminance Level (Guo et al., 2023) |
| DR | Dynamic Range: the log10 differences between the highest 2% luminance and the lowest 2% luminance. (Hu et al., 2022b) |

Table 19: Statistical analysis of data scenarios, time of day, and indoor/outdoor distribution.

| Motion Type | Environment | Time | | |
|---|---|---|---|---|
| | | Daylight | Twilight | Night |
| Local Motion | Indoor | 2016 | 1152 | 432 |
| | Outdoor | 2160 | 1440 | 1104 |
| Full Motion | Indoor | 3360 | 1920 | 720 |
| | Outdoor | 4272 | 3024 | 2400 |

## B  DATA EXAMPLES OF S2R-HDR

### B.1  MOTION MATERIALS

As demonstrated in Figure 14, the S2R-HDR dataset comprises three principal categories of motion materials: (a) human subjects with a comprehensive coverage of appearance variations, including garment diversity and gender attributes; (b) vehicular objects incorporating distinct transportation modalities with differential motion patterns; and (c) zoological specimens exhibiting biologically plausible locomotion characteristics. These motion materials are sourced from two origins: (1) manually created motion sequences by authors, and (2) pre-defined motion patterns from the Unreal Engine 5 materials we purchased, both of which are designed for integration into environmental contexts to facilitate dynamic motion synthesis.

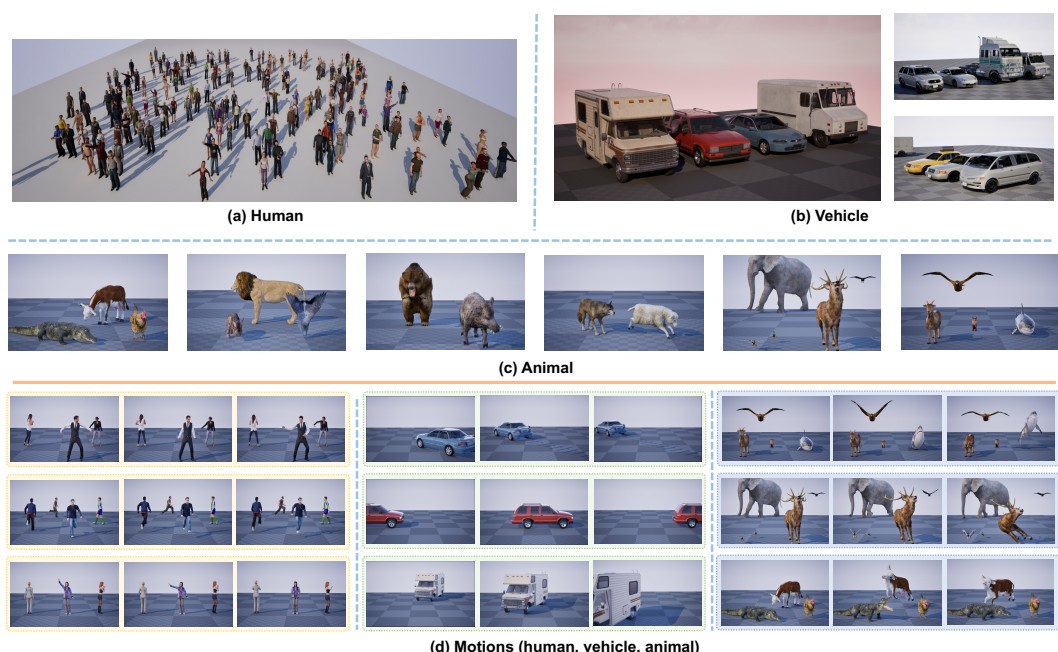

Figure 14: Illustration of motion materials.

### B.2  HIGH DYNAMIC RANGE ENVIRONMENTS

As illustrated in Figure 15, the S2R-HDR dataset presents a collection of high dynamic range environments encompassing both indoor and outdoor configurations. Through systematic utilization of Unreal Engine 5's Lumen global illumination system, we achieve precise control over environmental lighting parameters. This technical capability enables physics-based synthesis of illumination scenarios spanning three critical lighting regimes: daylight, twilight, and night.

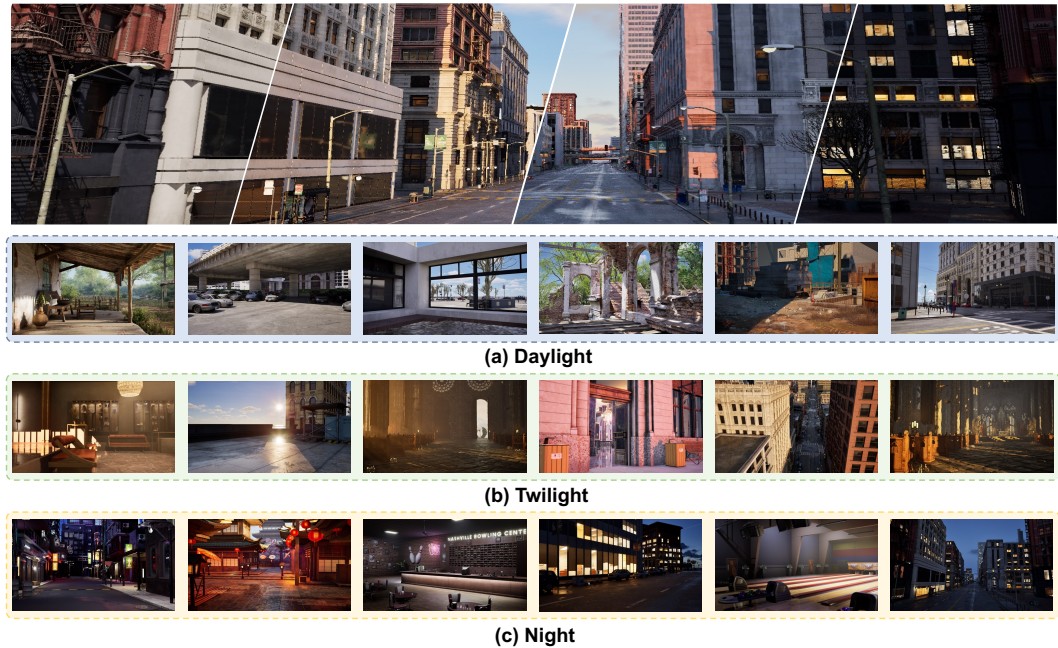

Figure 15: Illustration of high dynamic range environments.

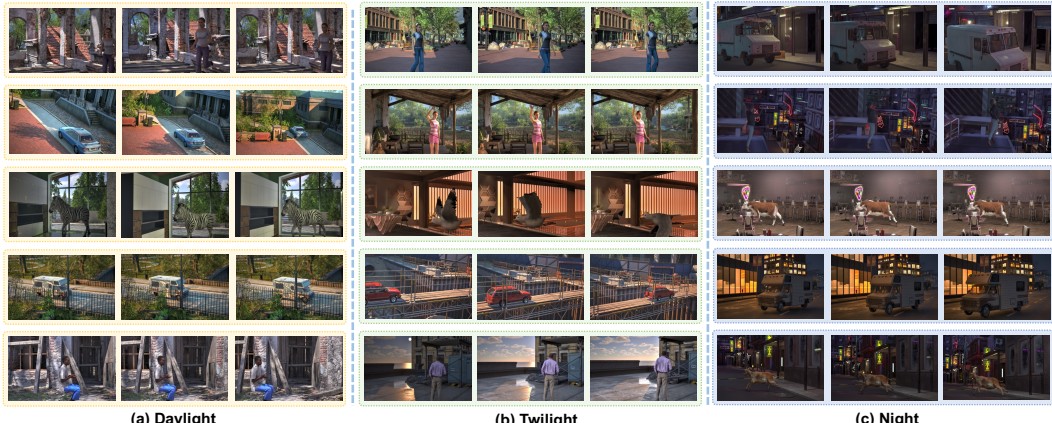

Figure 16: Illustration of image examples of our S2R-HDR.

## B.3 Synthesis of Camera Shake

To enhance the realism of our dataset and simulate inevitable device vibrations encountered in practical imaging scenarios, we introduce controlled camera motion perturbations in selected sequences. Specifically, 30% of the sequences incorporate Perlin noise-based jittering, applied simultaneously to both positional coordinates and rotational axes of the camera. The noise frequency and amplitude are adjusted to ensure perceptually plausible motion. This augmentation significantly improves the authenticity of the dataset while expanding its kinematic diversity, better approximating real-world camera operation.

## B.4 HDR Dataset Evaluation Metrics

To quantitatively assess the superiority of our dataset compared to real-world datasets, we employ seven evaluation metrics whose detailed definitions are provided in Table 18. Specifically, FHLP and EHL measure the extent of HDR. SI, CF and stdL quantify intra-frame diversity. ALL and DR evaluate overall style.

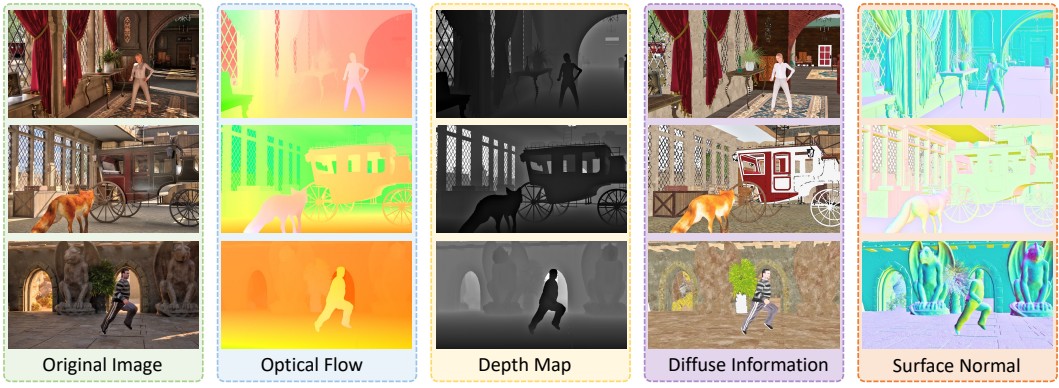

Figure 17: Different image types provided in S2R-HDR.

### B.5 SCENE AND MOTION DISTRIBUTIONS

Our dataset comprehensively encompasses diverse motion patterns, varied environments, and heterogeneous environmental illumination conditions. The distribution of different categories across the total collection of 24,000 images is detailed in Table 19.

### B.6 S2R-HDR IMAGE EXAMPLES

As shown in Figure 16, we present additional image examples of S2R-HDR.

### B.7 FURTHER APPLICATION OF S2R-HDR

Leveraging the advanced rendering capabilities of Unreal Engine 5, we have augmented our customized rendering pipeline with the capability to render multiple specialized image data types. Beyond producing standard output images, the system simultaneously generates four distinct auxiliary data modalities: optical flow fields, depth maps, diffuse reflectance information, and surface normal vectors. In other words, as shown in Figure 17, each frame has its corresponding four additional auxiliary information.

Currently, our dataset is still limited to the HDR fusion task. The provision of such comprehensive supplementary data aims to broaden the utility of S2R-HDR, facilitating its extension to diverse application domains beyond HDR imaging.

## C THE USE OF LARGE LANGUAGE MODELS

We only utilize LLMs to polish writing and correct grammar.

