# OpenReview forum: "S2R-HDR: A Large-Scale Rendered Dataset for HDR Fusion"
_ICLR.cc/2026/Conference — ICLR 2026 Poster_

### Official Review · Reviewer_cNjM · 2025-10-26

**Soundness:** 3
**Presentation:** 3
**Contribution:** 2
**Rating:** 6
**Confidence:** 5

**Summary:**

This paper proposes a method for generating image data in the field of HDR fusion using the Unreal Engine 5. To address the domain gap between synthetic and real data, it also introduces a domain adaptation approach applicable to both labeled and unlabeled datasets.

**Strengths:**

1. The arguments in the paper are well-grounded, and the writing is logically rigorous and easy to follow.


2. The proposed dataset offers a richer variety of HDR scenes, substantiated by partial visualizations and extensive quantitative comparisons. When combined with the strongly generalized domain adaptation method, it further enhances the performance in the field of HDR fusion.

**Weaknesses:**

The paper does not discuss the limitations of the proposed method. Is it effective in all scenarios? Please specify any potential constraints and provide examples of failure cases, which would help readers understand the method's boundaries.

**Questions:**

1. The authors emphasize the efficiency of their method. Could you provide the additional computational cost (for both training and inference) introduced by the domain adaptor? Specifically, during inference, if the method necessarily requires additional test-time-training from the adaptor for every new scenarios (e.g., day, night, dusk), the claimed efficiency might be compromised. Clarifying this practical overhead is crucial.


2. The LDR data in the Unreal Engine 5 synthetic dataset is generated via data augmentation, but the process (e.g., adding noise) is not detailed. For instance, the LDR examples in Figure 3 show no noise variation across exposures. This lack of realistic degradation might be a key reason for the poor generalization when trained solely on the proposed dataset. Enhancing the data synthesis pipeline with a more realistic degradation model could significantly boost the dataset's generalization capability and contribution.


3. Abundant HDR data from professional film-grade equipment can be acquired on the internet. What is the specific advantage of the dataset proposed in this work compared to these readily available sources? Please elaborate on its unique value.

---

> ### Author Response · Authors · 2025-11-21
>
> **W1**:Thank you for your suggestions. A limitation of our approach is that the proposed Adapter is primarily designed to mitigate the domain gap between synthetic and real data. Consequently, its ultimate effectiveness is inherently constrained by the capacity of the underlying base model (e.g., SAFNet). When the base network is unable to reliably handle highly challenging scenarios, such as those involving extremely large motions that preclude successful reconstruction, the Adapter cannot compensate for these fundamental shortcomings.
>
> **Q1**: We have already provided the test-time latency in Appendix A.10 TEST-TIME ADAPTATION RUNTIME, and we include the corresponding base model training time, standard testing time, and test-time adaptation time in the table below. During inference, our method only requires test-time adaptation (TTA) and does not involve any additional training. In addition, we would like to clarify that our paper does not make any claims regarding computational efficiency.
> | - | Training | Standard Testing|Test-time Adaptation|
> |-|-|-|-|
> | Runtime | 53h |0.3813s | 0.6128s |
> | PSNR-tested on SCT dataset | - | 43.85 | 47.23 |
>
> **Q2**: Thank you for pointing this out. Indeed, we did not include noise in Figure 3. However, we have described our noise synthesis strategy at different exposure levels in Appendix A.1 EXPERIMENT DETAILS and use the same degradation method as prior works [1, 2]. We will update Figure 3 accordingly in the final version. In future work, we plan to explore a more realistic data synthesis pipeline.
>
> [1] Xu, Gangwei, et al. "Hdrflow: Real-time hdr video reconstruction with large motions." Proceedings of the IEEE/CVF Conference on Computer Vision and Pattern Recognition. 2024.
>
> [2] Chen, Guanying, et al. "HDR video reconstruction: A coarse-to-fine network and a real-world benchmark dataset." Proceedings of the IEEE/CVF international conference on computer vision. 2021.
>
> **Q3**: Gathering data from YouTube remains extremely resource-intensive and time-consuming, and it also poses inherent risks of violating intellectual property rights. Moreover, HDR videos available on YouTube are typically captured using consumer or DSLR cameras, whose dynamic range is generally limited to around 15 stops (e.g., film-grade equipment | FX6 Full-frame Cinema Camera https://electronics.sony.com/imaging/cinema-line-cameras/all-cinema-line-cameras/p/ilmefx6v ). In contrast, a rendering-based pipeline can readily achieve over 30 stops of dynamic range (see
> https://forums.unrealengine.com/t/unreal-engine-dynamic-range/138047), providing significantly greater flexibility and fidelity for HDR data generation.
>
> Furthermore, we compute the Dynamic Range (DR) following the standard definition: the base-10 logarithmic difference between the highest 2% luminance and the lowest 2% luminance [1]. For example, a commonly used real-captured video
>  (https://www.ysjf.com/materials/m-000228550034727505920) yields a DR score of 2.2391, whereas our rendered HDR video
>  (https://anonymous.4open.science/r/AnonymousRespo-5F48/dataset_image/HDR/) achieves a DR score of 2.9631. This quantitatively demonstrates that our rendered data indeed provides a substantially higher dynamic range.
>
> [1] Xiangyu Hu, Liquan Shen, Mingxing Jiang, Ran Ma, and Ping An. LA-HDR: Light adaptive HDR reconstruction framework for single ldr image considering varied light conditions. IEEE Transactions on Multimedia, 25:4814–4829, 2022b.

---

> > ### Comment · Reviewer_cNjM · 2025-11-25
> >
> > I appreciate the authors' clarification. It has solidified my understanding that the core idea is to reframe the problem as one of domain adaptation through data generation.
> >
> > From the results shown, the method appears to be quite effective. However, I am uncertain about the upper bound of such a solution. My primary concern is that this approach may be a stopgap measure rather than a fundamental path toward completely resolving the issue.
> >
> > While I am grateful for the detailed response, I will conservatively stick with my initial rating for now.

---

> > > ### Author Response · Authors · 2025-11-25
> > >
> > > Thank you for the reviewer’s feedback and response. In many research areas, using simulated data to address data scarcity is a common approach. As demonstrated in the work of Chen et al. [1], they adopted a similar strategy to tackle related problems.
> > >
> > > [1] Chen, Suyi, et al. "Sira-pcr: Sim-to-real adaptation for 3d point cloud registration." Proceedings of the IEEE/CVF International Conference on Computer Vision, 2023.

---

### Official Review · Reviewer_Ug55 · 2025-10-28

**Soundness:** 3
**Presentation:** 2
**Contribution:** 3
**Rating:** 6
**Confidence:** 4

**Summary:**

The core contribution of this paper is addressing the generalization challenge in HDR fusion models caused by the scarcity of real-world data. By introducing the first large-scale synthetic dataset (S2R-HDR) containing 24,000 high-quality HDR samples rendered using Unreal Engine 5, which covers diverse dynamic scenes and lighting conditions, along with an innovative domain adaptation method (S2R-Adapter) that balances synthetic and real-world data knowledge through a dual-branch design, the model achieves state-of-the-art performance on real-world datasets. Through comprehensive experiments on real-world datasets, the combined use of S2R-HDR and S2R-Adapter enables models to achieve state-of-the-art performance, significantly reducing motion artifacts like ghosting and enhancing the restoration of details in extreme illumination conditions.

**Strengths:**

This study constructs a novel dataset for multi-exposure HDR fusion tasks, which contributes to the advancement of this research field. This dataset comprises diverse types of image data captured under various environmental conditions and exposure levels, demonstrating considerable diversity. Additionally, to address the discrepancy between synthetic and real data, this paper propose the S2R-Adapter as an effective solution for mitigating the domain gap between synthetic and real-world data. The two branches of this module are particularly noteworthy, as the shared branch and the transfer branch can effectively reduce the domain gap.This paper conducted both supervised and unsupervised experiments, and the experiments appear to be sufficiently comprehensive.

**Weaknesses:**

In the introduction, the author states that the dataset proposed by Barua et al. is designed for single-image LDR to HDR conversion tasks. However, this assertion appears to be inaccurate, as the dataset actually includes corresponding multi-exposure LDR images. In fact, it encompasses variations in color hues, saturation, exposure, and contrast levels. It is worth noting that this dataset is exceptionally large, containing 40K HDR image. Since this dataset is based on GTA, it inherently includes features such as vehicles, humans, and varying lighting conditions. The author only describes the practices of shared branches and transfer branches, but throughout the entire paper, I cannot fully comprehend how these two branches achieve their intended functions. What is the theoretical basis for this? Or is it solely validated through experimental results? In addition, it appears that the author has omitted the UE5 dataset construction flowchart.

**Questions:**

Q1: Is the first claim accurate? The dataset compiled by Barua et al. contains 40,000 HDR images, resulting in a total dataset size of 1M. The volume of this data significantly exceeds that of your dataset.
Q2:Why is the dataset from Barua et al. unsuitable for multi-exposure HDR image reconstruction? The author should revise the introduction section regarding the description of the Barua et al. dataset, preferably providing a detailed comparison between this dataset and the proposed dataset in the paper. The dataset from Barua et al. includes multi-exposure data.
Q3: The authors need to elaborate on the specific rendering process using UE5, including scene creation and virtual camera capture. Could a corresponding flowchart be provided? If acceptable, could the process script also be included?
Q4: Why can the shared branch be utilized to preserve knowledge for rendering data? Why can the transfer branch learn domain-specific knowledge? The authors do not appear to provide detailed theoretical proofs, offering only some explanations in the experimental section. I could not find the basis for the corresponding effects of these two modules in the paper.

---

> ### Author Response · Authors · 2025-11-21
>
> **Q1Q2**:  In the work of Barua et al., 2025, the method primarily focuses on single-LDR to HDR reconstruction and additionally supoports multi-exposure, but without considering motion or dynamic scenes (we also confirmed this with the authors via email). However, real-world capture scenarios are predominantly dynamic, and motion introduces substantial additional challenges. For example, the SCT dataset contains exclusively dynamic HDR reconstruction scenes.
>
> Moreover, our dataset provides a substantially higher spatial resolution—1920 × 1080 compared to 1024 × 1024 in their work. Beyond resolution, we further supply rich auxiliary modalities, including optical flow, depth, diffuse, and normal maps, which extend the dataset’s applicability to a broader range of vision tasks. Finally, we introduce an Adapter module to explicitly bridge the domain gap between synthetic and real data, thereby improving model transferability and real-world robustness.
>
> **Q3**: Thank you for the suggestion. We now provide a complete and transparent overview of our dataset-generation pipeline in the link https://anonymous.4open.science/r/AnonymousRespo-5F48/Flowchart.svg. Because this pipeline allows us to directly control the lighting parameters of the sources in the UE5 simulation environment (including their positions, intensities, etc.), we can synthesize a highly diverse set of illumination conditions. In addition, we can manipulate the pose and motion of objects in the scene by binding different animation sequences to them, thereby producing a wide variety of motion patterns. With the exception of the use of several high-quality UE5 scene assets that we purchased for scene construction, the rest of the pipeline is fully reproducible.
>
> **Q4**: We provide empirical validation for the distinct roles of our branches through the t-SNE visualization in Figure 4 (described in Sec. 4, Lines 251-258). In this analysis, we visualize the feature distributions extracted from each branch when the model is fed with both rendered and real images. The results are as follows:
> - Shared Branch: Features extracted from the shared branch show highly consistent and overlapping distributions for both the real and rendered domains. This indicates that the shared branch successfully learns to ignore the domain gap, thereby preserving the core, domain-invariant knowledge from the source data and preventing catastrophic forgetting.
> - Transfer Branch: In contrast, the transfer branch effectively separates the features from the real and rendered domains into two distinct clusters. This demonstrates its capability to model domain-specific characteristics, allowing it to better capture the real data distribution and learn new knowledge specific to the target (real) domain.
>
> This visualization provides direct empirical evidence for our design, confirming that the branches function as intended to balance knowledge preservation and domain adaptation.

---

### Official Review · Reviewer_2u9g · 2025-10-31

**Soundness:** 3
**Presentation:** 3
**Contribution:** 3
**Rating:** 6
**Confidence:** 5

**Summary:**

This paper mainly proposes a large-scale synthetic HDR dataset and a post-processing module for domain adaptation between the synthetic and real-world domains. Specifically, a large-scale HDR dataset is essential for training powerful HDR models, especially in the era of large-scale generative models. Compared with existing datasets, the proposed HDR dataset leverages Unreal Engine 5 to synthesize data, resulting in 24,000 high-quality HDR frames featuring diverse motion types, lighting conditions, and both indoor and outdoor scenes. Current HDR models trained on small real datasets often struggle with large-scale motion and diverse lighting conditions, whereas models trained on the proposed large-scale dataset achieve significantly better performance.

Regarding the proposed post-processing module, called S2R-Adapter, it aims to reduce the domain gap between synthetic and real-world data. This module can be used in two modes: 1). fine-tuning mode, which requires labeled datasets for supervised training; 2). self-supervised mode (or test-time mode), which can be applied to unlabeled data without ground-truth images.

Experimental results demonstrate that the proposed large-scale synthetic HDR dataset effectively improves the performance of existing HDR fusion models. They also show that the proposed dataset and adapter achieve better generalization across different datasets.

Overall, the quality of this paper is above the borderline and it is likely to be accepted. However, several questions and concerns remain are  listed in below sessions.

**Strengths:**

1. The motivation of this paper is very clear, which aims to tackles the dataset bottleneck in HDE imaging based on multiple-image fusion.
2.  It also point out the challenging issue regarding to domain gap between the synthesized and real-world data, and propose S2R-adapter to handle this issue.
3. The paper demonstrates comprehensive experiment results, which are promising.

**Weaknesses:**

1. One of the major contributions of this paper is the proposed large-scale dataset, but the paper does not clearly describe how this dataset is created. For example, it only mentions that Unreal Engine 5 is used with varying tone-mapping and gamma-correction parameters to render different data, but the explicit pipeline for dataset generation is missing. Why does this synthetic pipeline effectively produce data with diverse lighting conditions and object motions? Is this dataset curation pipeline reproducible?

2. In the S2R-Adapter framework, the paper shows that it uses a diffusion model as the backbone. Which specific diffusion model is used in the experiments? If the S2R-Adapter is employed, what is the resulting model complexity and inference time? During fine-tuning, how large a dataset is required to train the adapter effectively? It would be better if the experimental settings elaborated more on the training details.

3. The S2R-Adapter can be used in self-supervised mode, and the paper mentions that “we dynamically adjust the scale factors using domain shift.” It is unclear how the domain shift is measured and how the model determines whether the domain shift is large or small. The paper refers to data augmentation and uncertainty estimation — but why does this strategy effectively measure domain shift and determine its magnitude?

4. In Table 2, it would be better to include the model complexity and inference time for all compared models. Additionally, how many adaptation steps does the proposed S2R-Adapter use in the experiments? For the self-supervised mode, how does the performance vary across different datasets?

**Questions:**

I jointly discuss the weakness and questions on this paper in the Weakness section. The authors can prepare their rebuttal referred to the comments in the Weakness section.

---

> ### Author Response · Authors · 2025-11-21
>
> **W1**: Thank you for the suggestion. We now provide a complete and transparent overview of our dataset-generation pipeline in the link https://anonymous.4open.science/r/AnonymousRespo-5F48/Flowchart.svg. Because this pipeline allows us to directly control the lighting parameters of the sources in the UE5 simulation environment (including their positions, intensities, etc.), we can synthesize a highly diverse set of illumination conditions. In addition, we can manipulate the pose and motion of objects in the scene by binding different animation sequences to them, thereby producing a wide variety of motion patterns. With the exception of the use of several high-quality UE5 scene assets that we purchased for scene construction, the rest of the pipeline is fully reproducible.
>
> **W2**: First, we clarify that no diffusion model is used in our framework. As stated in Sec. 5.1 (Line 353), we selected SCTNet (a Transformer-based method) and SAFNet (a CNN-based method) as our backbones. The complexity overhead from the S2R-Adapter is minimal during training. Crucially, as mentioned in Line 89, the adapter is merged into the backbone via reparameterization for inference, which incurs no extra computational overhead. For fine-tuning, we used the same datasets for all methods to ensure a fair comparison, specifically the SCT dataset with 108 training samples and the Challenge123 dataset with 96 samples (Sec. 5, Line 307).
>
> **W3**: As illustrated in Sec. 4 (Lines 291-295), we measure domain shift using model uncertainty. This approach is a well-established practice, following prior works such as Wang et al. (2022) and Liu et al. (2023a). In our method, we augment each input sample N times (e.g., adjusting exposure, white balance, noise, and applying random flips) and calculate the variance across the N outputs. This variance serves as the uncertainty value U(x). The effectiveness of this strategy is confirmed by our experimental results; as shown in the ablation study in Tab. 6, using uncertainty to dynamically adjust the adapter's scale factors enables better control of knowledge transfer across varying domain shifts.
>
> **W4**: Thank you for the suggestion. We will add model complexity and inference time to Table 2 in the revision. Regarding adaptation steps, for supervised adaptation with ground-truth labels, we fine-tune the model for about 30 epochs. For the self-supervised mode (test-time adaptation without ground-truth), the model adapts as it processes the data during test time, so each sample is seen only once. The inference time can be found in Appendix A.10 TEST-TIME ADAPTATION RUNTIME. A full performance comparison of this self-supervised mode across various datasets can be found in Table 3, with a detailed description in Sec. 5 (Lines 365-409).

---

### Official Review · Reviewer_rtUP · 2025-11-01

**Soundness:** 2
**Presentation:** 2
**Contribution:** 3
**Rating:** 4
**Confidence:** 3

**Summary:**

This paper proposes S2R-HDR, a synthetic dataset of 24,000 HDR images rendered using Unreal Engine 5, representing a large-scale increase over existing datasets. To bridge the synthetic-to-real domain gap, they introduce S2R-Adapter, a dual-branch domain adaptation module with shared and transfer branches. The method demonstrates state-of-the-art performance on the reported benchmarks through both supervised fine-tuning and test-time adaptation

**Strengths:**

- The major strength of the dataset is definitely its scale.
- The dataset presents a high variety of environments as opposed to real captures that are often in the same environment, or provide a very small number of examples in various environments.
- The idea of using the adapter for domain adaptation is sound and seems to be working well.
- The quantitative and qualitative results look convincing.

**Weaknesses:**

- The authors introduce a shake simulation into camera poses. It would be great to assess whether those are close to the real world. In a fairly easy way, one could record sensor data with a phone capture to compare the two (potentially real patterns could also be transferred to the sim).
- The authors report the size of the other dataset considering sequences of images, while the presented dataset is reported as the number of frames, which is misleading to the reader.
- It would be good to see high-resolution images from the dataset; the compressed ones in the paper make it harder to judge quality.
- I believe there is room for additional experiments to help with assessing the weight of the contributions. In Table 4, we see that training only on S2R-HDR is typically not enough but rather needs fine-tuning on the target dataset. It raises the question whether the scale or data diversity is needed - one could add all datasets to the mix in the comparison. Further, and more importantly, what happens if we take one or more of the real-world datasets to train the approach and then use the adapter on the target data? This would clarify the impact of the dataset a lot.
- Continuing the argument regarding experimentation with the source data, part of the issues with the real-world dataset is imperfect real ISPs versus the synthetic pipeline. Would the adapter approach work, for e.g. a collection of real-world data (e.g. from YouTube) with a synthetic degradation applied?
- It seems that the approach does not work on the Kalantari dataset - Table 11 misses the entry of SAFNet trained on Kalantari (SAFNet reports higher values). It would be better to comment on that instead of skipping completely. Similarly, in Table 10, the authors report 44.13dB on Kalantari, whereas the SCT paper reports 44.49dB.

**Questions:**

- What are the benefits of your approach over the one from (Barua et al., 2025), i.e. could their approach be adjusted to multiple exposures?
- What kind of features were extracted for t-SNE?
- Some more statistics on the dataset would be useful, e.g. more analysis on motion (quantifying local and global by presenting optical flow analysis), and some analysis on the number of different environments.

---

> ### Author Response · Authors · 2025-11-21
>
> **W1**: Thank you for your insightful suggestions. However, comparing the jitter poses recorded from mobile phone sensors with simulated jitter is a highly challenging task. The actual jitter is a highly random and uncertain process, making it difficult to achieve perfect alignment between the two. What we can do is capture the characteristics of real camera jitter and then attempt to simulate this process, striving for consistency with its paradigm. Currently, it is widely recognized in the rendering community that the motion trajectory of real camera shake is a temporally continuous random process, primarily dominated by low-frequency components with a small amount of high-frequency detail, which can be effectively modeled using Perlin noise. We therefore adopt Perlin noise to simulate camera shake in real-world image capture. Perlin noise has the following desirable properties: (i) it is a smooth random function that is continuous and differentiable; (ii) the power spectrum of hand-held camera shake is dominated by low-frequency components with a small amount of high-frequency jitter, which closely matches the inherent multi-scale frequency structure of Perlin noise and can be controlled via its octaves; and (iii) Perlin noise allows precise control over amplitude, frequency, and temporal speed, while being both realistic and exactly reproducible. These properties make Perlin noise particularly suitable for simulating camera shake in synthetic environments.
>
> **W2**: Thank you for pointing this out. We will explicitly claim this in the final version.
>
> **W3**:  We provide high-quality raw HDR data (dataset_image/HDR), which can be visualized using software such as Photomatix. The dataset can be accessed at: https://anonymous.4open.science/r/AnonymousRespo-5F48.
>
>  **W4**: In terms of data effectiveness, our main idea lies in the fact that there are two fundamental limitations of real-world datasets focused on HDR reconstruction: (1) Scalability constraints (laborious acquisition and limited data quantity), (2) Absence of ground-truth labels (inherent to physical capture limitations). These issues critically hinder network training yet remain challenging to resolve through real-data collection. By contrast, synthetic datasets inherently resolve these two limitations.  There is indeed a domain gap between synthetic and real data, which is an issue inevitably introduced by synthetic methods. However, compared to the challenges inherent in acquiring real data, the domain gap problem is more tractable through our S2R-Adapter framework. In summary, we shift the challenge from the intractable constraints of real-data collection to a solvable domain adaptation problem. Within this framework, when a small amount of real labeled data is available, a light finetuning stage is sufficient to reduce the domain discrepancy. When no real labeled data is available, our Adapter module enables effective test-time adaptation (TTA) to alleviate the domain gap.
>
> For the mixed-data experiments, we conduct two comparisons with SAFNet: (1) joint training on both the SCT and Challenge123 datasets; (2) training on Challenge123 and then adapting to SCT using the Adapter. As shown in the following table, the experimental results demonstrate that our dataset and approach achieve the best overall performance.
>
> | Training| PNSR-$\mu$ | PNSR-$\ell$ | SSIM-$\mu$ | SSIM-$\ell$ |
> |-|-|-|-|-|
> | SCT | 42.66 | 48.38 | 0.9831 | 0.9955 |
> | Challeng123 | 35.14 | 38.77 | 0.9619 | 0.9868 |
> | SCT&Challenge | 42.12 | 45.14 | 0.9853 | 0.9941 |
> | S2R-HDR with S2R-Adapter on SCT | **43.44** | **48.90** | **0.9864** | **0.9959** |
> | Challenge with S2R-Adapter on SCT | 42.92 | 47.05 | 0.9856 | 0.9951 |

---

> > ### Author Response · Authors · 2025-11-21
> >
> > **W5**:  Gathering data from YouTube remains extremely resource-intensive and time-consuming, and it also poses inherent risks of violating intellectual property rights. Moreover, HDR videos available on YouTube are typically captured using consumer or DSLR cameras, whose dynamic range is generally limited to around 15 stops (e.g., film-grade equipment | FX6 Full-frame Cinema Camera https://electronics.sony.com/imaging/cinema-line-cameras/all-cinema-line-cameras/p/ilmefx6v ). In contrast, a rendering-based pipeline can readily achieve over 30 stops of dynamic range (see
> > https://forums.unrealengine.com/t/unreal-engine-dynamic-range/138047), providing significantly greater flexibility and fidelity for HDR data generation.
> > Furthermore, we compute the Dynamic Range (DR) following the standard definition: the base-10 logarithmic difference between the highest 2% luminance and the lowest 2% luminance [1]. For example, a commonly used real-captured video (https://www.ysjf.com/materials/m-000228550034727505920) yields a DR score of 2.2391, whereas our rendered HDR video (https://anonymous.4open.science/r/AnonymousRespo-5F48/dataset_image/HDR/) achieves a DR score of 2.9631. This quantitatively demonstrates that our rendered data indeed provides a substantially higher dynamic range.
> >
> >  [1] Xiangyu Hu, Liquan Shen, Mingxing Jiang, Ran Ma, and Ping An. LA-HDR: Light adaptive HDR reconstruction framework for single ldr image considering varied light conditions. IEEE Transactions on Multimedia, 25:4814–4829, 2022b.
> >
> > **W6**:  In Table 11, our SAFNet trained on S2R-HDR with the S2R-Adapter has never seen the Kalantari training data; its performance is obtained purely through test-time adaptation. Therefore, directly comparing this result against a model that is explicitly trained on the Kalantari dataset is inherently unfair. For the SCT results in Table 10, we used the official implementation code to re-train the baseline. Our reproduced results are consistent with those reported by other researchers(https://github.com/Zongwei97/SCTNet/issues/5).
> >
> > **Q1**:   In the work of Barua et al., 2025, the method primarily focuses on single-LDR to HDR reconstruction and additionally supoports multi-exposure, but without considering motion or dynamic scenes (we also confirmed this with the authors via email). However, real-world capture scenarios are predominantly dynamic, and motion introduces substantial additional challenges. For example, the SCT dataset contains exclusively dynamic HDR reconstruction scenes.
> >
> > Moreover, our dataset provides a substantially higher spatial resolution—1920 × 1080 compared to 1024 × 1024 in their work. Beyond resolution, we further supply rich auxiliary modalities, including optical flow, depth, diffuse, and normal maps, which extend the dataset’s applicability to a broader range of vision tasks. Finally, we introduce an Adapter module to explicitly bridge the domain gap between synthetic and real data, thereby improving model transferability and real-world robustness.
> >
> > **Q2**: Please refer to Appendix B.4: HDR Dataset Evaluation Metrics.
> >
> > **Q3**: We thank you for pointing out that our previous analysis was not sufficiently comprehensive. We now provide the following additional scene-level analysis. First, following the same methodology as in [1], we analyze the directions of motion in the optical flow. The image(https://anonymous.4open.science/r/AnonymousRespo-5F48/flow.png )reports the proportion of pixels corresponding to different motion directions in the dataset, from which it can be observed that the motion patterns in our dataset are highly diverse. Second, Table 18 summarizes the distribution of illumination conditions and locations. In terms of scene semantics, our dataset covers a wide variety of environments, including home interiors, urban streets, churches, rural areas, parks, subway stations, workshops, and more. Regarding the foreground assets, Fig. 12 further illustrates the richness of our foreground content together with the associated motion patterns.
> >
> > [1] Shu, Yong, et al. "Towards real-world HDR video reconstruction: A large-scale benchmark dataset and a two-stage alignment network." Proceedings of the IEEE/CVF Conference on Computer Vision and Pattern Recognition. 2024.

---

### Meta-Review · Area_Chair_r7AE · 2025-12-24

**Summary:**

This paper receives three marginally above the acceptance threshold and one marginally below the acceptance threshold. After carefully checking the rebuttal, most of the issues are addressed, including those raised by rtUP. As a result, this paper can be accepted.

**Reviewer Scores:**

NA

---

### Decision · Program_Chairs · 2026-01-26

Accept (Poster)